# On cross-frequency phase-phase coupling between theta and gamma oscillations in the hippocampus

Robson Scheffer-Teixeira, Adriano BL Tort*

Brain Institute, Federal University of Rio Grande do Norte, Natal, Brazil

**Abstract** Phase-amplitude coupling between theta and multiple gamma sub-bands is a hallmark of hippocampal activity and believed to take part in information routing. More recently, theta and gamma oscillations were also reported to exhibit phase-phase coupling, or n:m phase-locking, suggesting an important mechanism of neuronal coding that has long received theoretical support. However, by analyzing simulated and actual LFPs, here we question the existence of theta-gamma phase-phase coupling in the rat hippocampus. We show that the quasi-linear phase shifts introduced by filtering lead to spurious coupling levels in both white noise and hippocampal LFPs, which highly depend on epoch length, and that significant coupling may be falsely detected when employing improper surrogate methods. We also show that waveform asymmetry and frequency harmonics may generate artifactual n:m phase-locking. Studies investigating phase-phase coupling should rely on appropriate statistical controls and be aware of confounding factors; otherwise, they could easily fall into analysis pitfalls.

*For correspondence: tort@neuro.ufrn.br

**Competing interests:** The authors declare that no competing interests exist.

## Introduction

Local field potentials (LFPs) exhibit oscillations of different frequencies, which may co-occur and also interact with one another (*Jensen and Colgin, 2007*; *Tort et al., 2010*; *Hyafil et al., 2015*). Cross-frequency phase-amplitude coupling between theta and gamma oscillations has been well described in the hippocampus, whereby the instantaneous amplitude of gamma oscillations depends on the instantaneous phase of theta (*Scheffer-Teixeira et al., 2012*; *Schomburg et al., 2014*). More recently, hippocampal theta and gamma oscillations were also reported to exhibit n:m phase-phase coupling, in which multiple gamma cycles are consistently entrained within one cycle of theta (*Belluscio et al., 2012*; *Zheng and Zhang, 2013*; *Xu et al., 2013*, *2015*; *Zheng et al., 2016*). The existence of different types of cross-frequency coupling suggests that the brain may use different coding strategies to transfer multiplexed information.

Coherent oscillations are believed to take part in network communication by allowing opportunity windows for the exchange of information (*Varela et al., 2001*; *Fries, 2005*). Standard phase coherence measures the constancy of the phase difference between two oscillations of the same frequency (*Lachaux et al., 1999*; *Hurtado et al., 2004*), and has been associated with cognitive processes such as decision-making (*DeCoteau et al., 2007*; *Montgomery and Buzsáki, 2007*; *Nácher et al., 2013*). Similarly to coherence, cross-frequency phase–phase coupling, or n:m phase-locking, also relies on assessing the constancy of the difference between two phase time series (*Tass et al., 1998*). However, in this case the original phase time series are accelerated, so that their instantaneous frequencies can match. Formally, n:m phase-locking occurs when $\Delta\varphi_{nm}(t) = n * \varphi_B(t) - m * \varphi_A(t)$ is non-uniform but centered around a preferred value, where $n * \varphi_B$ ($m * \varphi_A$) denotes the phase of oscillation B (A) accelerated n (m) times (*Tass et al., 1998*). For example, the instantaneous phase of theta oscillations at 8 Hz needs to be accelerated five times to match in

**eLife digest** Neuroscientists have long sought to understand how the brain works by analyzing its electrical activity. Placing electrodes on the scalp or lowering them into the brain itself reveals rhythmic waves of activity known as oscillations. These arise when large numbers of neurons fire in synchrony. Recordings reveal that the frequency of these oscillations – the number of cycles of a wave per second, measured in Hertz – can vary between brain regions, and within a single region over time. Moreover, oscillations with different frequencies can co-exist and interact with one another.

Within the hippocampus, an area of the brain involved in memory, two types of oscillations dominate: theta waves and gamma waves. Theta waves are relatively slow waves, with a frequency between 5 and 10 Hertz. Gamma waves are faster, with a frequency of up to 100 Hertz. Recent work has suggested that gamma waves and theta waves show a phenomenon called phase-phase coupling. Since gamma waves are faster than theta waves, multiple cycles of gamma can occur during a single cycle of theta. Phase-phase coupling is the idea that gamma and theta waves align themselves, such that gamma waves always begin at the same relative position within a theta wave. This was thought to help the hippocampus to encode memories.

Using computer simulations and recordings from the rat hippocampus, Scheffer-Teixeira and Tort have now reexamined the evidence for theta-gamma phase-phase coupling. The new results suggest that previous reports describing the phenomenon may have relied on inadequate statistical techniques. Using stringent control analyses, Scheffer-Teixeira and Tort find no evidence for prominent theta-gamma phase-phase coupling in the hippocampus. Instead, the simulations suggest that what appeared to be statistically significant coupling may in reality be an artifact of the previous analysis.

Phase-phase coupling of theta and gamma waves has also been reported in the human hippocampus. The next step therefore is to apply these more robust analysis techniques to data from the human brain. While revisiting previously accepted findings may not always be popular, it will likely be essential if neuroscientists want to accurately understand how new memories are formed.

frequency a 40 Hz gamma. A 1:5 phase-phase coupling is then said to occur if theta accelerated five times has a preferred phase lag (i.e., a non-uniform phase difference) in relation to gamma; or, in other words, if five gamma cycles have a consistent phase relationship to one theta cycle.

Cross-frequency phase-phase coupling has previously been hypothesized to take part in memory processes (*Lisman and Idiart, 1995*; *Jensen and Lisman, 2005*; *Lisman, 2005*; *Schack and Weiss, 2005*; *Sauseng et al., 2008*, *2009*; *Holz et al., 2010*; *Fell and Axmacher, 2011*). Recent findings suggest that the hippocampus indeed uses such a mechanism (*Belluscio et al., 2012*; *Zheng and Zhang, 2013*; *Xu et al., 2013*, *2015*; *Zheng et al., 2016*). However, by analyzing simulated and actual hippocampal LFPs, in the present work we question the existence of theta-gamma phase-phase coupling.

## Results

### Measuring n:m phase-locking

We first certified that we could reliably detect n:m phase-locking when present. To that end, we simulated a system of two Kuramoto oscillators – a 'theta' and a 'gamma' oscillator – exhibiting variability in instantaneous frequency (see Materials and methods). The mean natural frequency of the theta oscillator was set to 8 Hz, while the mean natural frequency of the gamma oscillator was set to 43 Hz (*Figure 1A*). When coupled, the mean frequencies aligned to a 1:5 factor by changing to 8.5 Hz and 42.5 Hz, respectively (see *Guevara and Glass, 1982*; *García-Alvarez et al., 2008*; *Canavier et al., 2009*). *Figure 1B* depicts three versions of accelerated theta phases (m = 3, 5 and 7) along with the instantaneous gamma phase (n = 1) of the coupled oscillators (see *Figure 1—figure supplement 1* for the uncoupled case). Also shown are the time series of the difference between gamma and

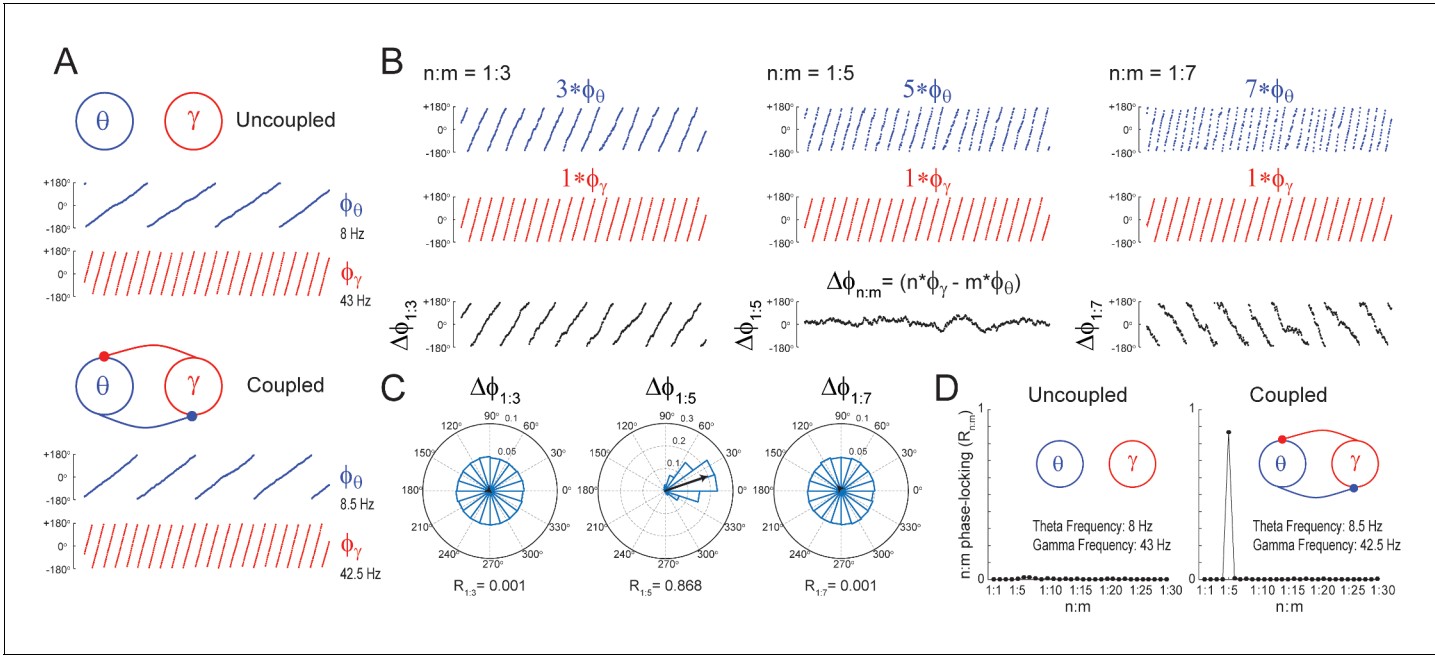

**Figure 1.** Measuring cross-frequency phase-phase coupling. (A) Traces show 500 ms of the instantaneous phase time series of two Kuramoto oscillators (see Materials and methods). When uncoupled (top panels), the mean natural frequencies of the 'theta' and 'gamma' oscillator are 8 Hz (blue) and 43 Hz (red), respectively. When coupled (bottom panels), the oscillators have mean frequencies of 8.5 Hz and 42.5 Hz. (B) Top blue traces show the instantaneous phase of the coupled theta oscillator for the same period as in A but accelerated *m* times, where m = 3 (left), 5 (middle) and 7 (right). Middle red traces reproduce the instantaneous phase of the coupled gamma oscillator (i.e., n = 1). Bottom black traces show the instantaneous phase difference between gamma and accelerated theta phases ($\Delta\varphi_{nm}$). Notice roughly constant $\Delta\varphi_{nm}$ only when theta is accelerated m = 5 times, which indicates 1:5 phase-locking. See *Figure 1—figure supplement 1* for the uncoupled case. (C) $\Delta\varphi_{nm}$ distributions for the coupled case (epoch length = 100 s). Notice uniform distributions for n:m = 1:3 and 1:7, and a highly concentrated distribution for n:m = 1:5. The black arrow represents the mean resultant vector for each case (see Materials and methods). The length of this vector ($R_{n:m}$) measures the level of n:m phase-locking. See *Figure 1—figure supplement 1* for the uncoupled case. (D) Phase-locking levels for a range of n:m ratios for the uncoupled (left) and coupled (right) oscillators (epoch length = 100 s).

The following figure supplement is available for figure 1:

**Figure supplement 1.** Uncoupled oscillators display uniform $\Delta\varphi_{nm}$ distribution.

accelerated theta phases ($\Delta\varphi_{nm}$). The instantaneous phase difference has a preferred lag only for m = 5; when m = 3 or 7, $\Delta\varphi_{nm}$ changes over time, precessing forwards (m = 3) or backwards (m = 7) at an average rate of 17 Hz. Consequently, $\Delta\varphi_{nm}$ distribution is uniform over 0 and $2\pi$ for m = 3 or 7, but highly concentrated for m = 5 (*Figure 1C*). The concentration (or 'constancy') of the phase difference distribution is used as a metric of n:m phase-locking. This metric is defined as the length of the mean resultant vector ($R_{n:m}$) over unitary vectors whose angle is the instantaneous phase difference ($e^{i\Delta\varphi_{nm}(t)}$), and thereby it varies between 0 and 1. For any pair of phase time series, an $R_{n:m}$ 'curve' can be calculated by varying m for n = 1 fixed. As shown in *Figure 1D*, the coupled – but not uncoupled – oscillators exhibited a prominent peak for n:m = 1:5, which shows that $R_{n:m}$ successfully detects n:m phase-locking.

## Filtering-induced n:m phase-locking in white noise

We next analyzed white-noise signals, in which by definition there is no structured activity; in particular, the spectrum is flat and there is no true n:m phase-locking. $R_{n:m}$ values measured from white noise should be regarded as chance levels. We band-pass filtered white-noise signals to extract the instantaneous phase of theta (θ: 4–12 Hz) and of multiple gamma bands (*Figure 2A*): slow gamma ($\gamma_S$: 30–50 Hz), middle gamma ($\gamma_M$: 50–90 Hz), and fast gamma ($\gamma_F$: 90–150 Hz). For each frequency pair, we constructed n:m phase-locking curves for epochs of 1 and 10 s, with n = 1 fixed and m

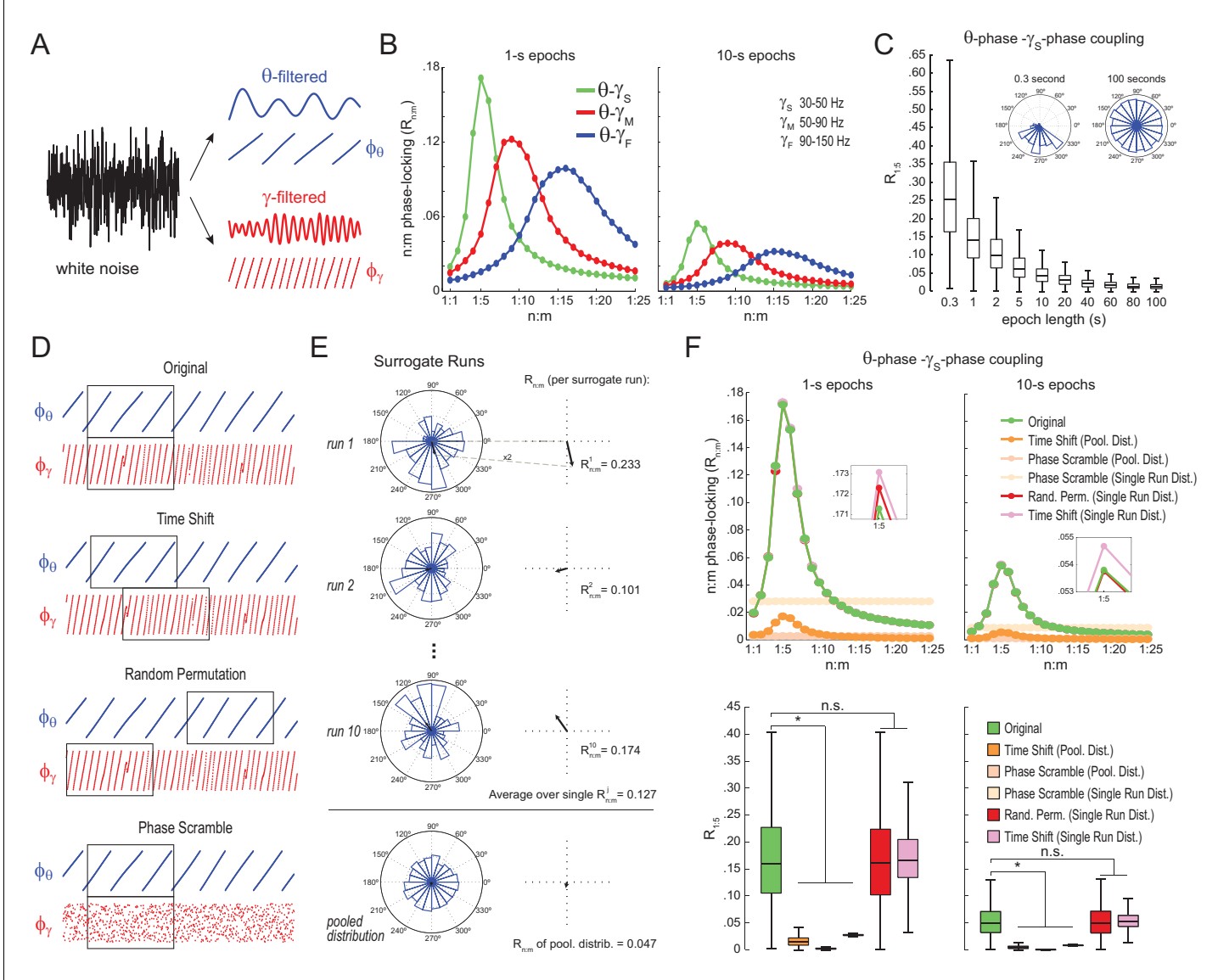

**Figure 2.** Detection of spurious n:m phase-locking in white-noise signals due to inappropriate surrogate-based statistical testing. (**A**) Example white-noise signal (black) along with its theta- (blue) and gamma- (red) filtered components. The corresponding instantaneous phases are also shown. (**B**) n:m phase-locking levels for 1- (left) and 10 s (right) epochs, computed for noise filtered at theta (θ; 4–12 Hz) and at three gamma bands: slow gamma (γ$_S$; 30–50 Hz), middle gamma (γ$_M$; 50–90 Hz) and fast gamma (γ$_F$; 90–150 Hz). Notice R$_{n:m}$ peaks in each case. (**C**) Boxplot distributions of θ−γ$_S$ R$_{1:5}$ values for different epoch lengths (n = 2100 simulations per epoch length). The inset shows representative Δφ$_{nm}$ distributions for 0.3- and 100 s epochs. (**D**) Overview of surrogate techniques. See text for details. (**E**) Top panels show representative Δφ$_{nm}$ distributions for single surrogate runs (*Time Shift*; 10 runs of 1 s epochs), along with the corresponding R$_{n:m}$ values. The bottom panel shows the pooled Δφ$_{nm}$ distribution; the R$_{n:m}$ of the pooled distribution is lower than the R$_{n:m}$ of single runs (compare with values for 1- and 10 s epochs in panel **C**). (**F**) Top, n:m phase-locking levels computed for 1- (left) or 10 s (right) epochs using either the *Original* or five surrogate methods (insets are a zoomed view of R$_{n:m}$ peaks). Bottom, R$_{1:5}$ values for white noise filtered at θ and γ$_S$. Original R$_{n:m}$ values are not different from R$_{n:m}$ values obtained from single surrogate runs of *Random Permutation* and *Time Shift* procedures. Less conservative surrogate techniques provide lower R$_{n:m}$ values and lead to the spurious detection of θ−γ$_S$ phase-phase coupling in white noise. *p<0.01, n = 2100 per distribution, one-way ANOVA with Bonferroni post-hoc test.

The following figure supplements are available for figure 2:

**Figure supplement 1.** Filtering induces quasi-linear phase shifts in white-noise signals.

**Figure supplement 2.** Filter bandwidth influences n:m phase-locking levels in white-noise signals.

**Figure supplement 3.** Uniform p-value distributions upon multiple testing of *Original* R$_{n:m}$ values against *Single Run* R$_{n:m}$ surrogates.

varying from 1 to 25 (*Figure 2B*). In each case, phase-phase coupling was high within the ratio of the analyzed frequency ranges: $R_{n:m}$ peaked at m = 4–6 for $\theta-\gamma_S$, at m = 7–11 for $\theta-\gamma_M$, and at m = 12–20 for $\theta-\gamma_F$. Therefore, the existence of a 'bump' in the $R_{n:m}$ curve may merely reflect the ratio of the filtered bands and should not be considered as evidence for cross-frequency phase-phase coupling: even filtered white-noise signals exhibit such a pattern.

The bump in the $R_{n:m}$ curve of filtered white noise is explained by the fact that neighboring data points are not independent. In fact, the phase shift between two consecutive data points follows a probability distribution highly concentrated around $2*\pi*f_c*dt$, where $f_c$ is the filter center frequency and dt the sampling period (*Figure 2—figure supplement 1*). For instance, for dt = 1 ms (1000 Hz sampling rate), consecutive samples of white noise filtered between 4 and 12 Hz are likely to exhibit phase difference of 0.05 rad (8 Hz center frequency); likewise, signals filtered between 30 and 50 Hz are likely to exhibit phase differences of 0.25 rad (40 Hz center frequency). In turn, the 'sinusoidality' imposed by filtering leads to non-zero $R_{n:m}$ values, which peak at the ratio of the center frequencies, akin to the fact that perfect 8 Hz and 40 Hz sine waves have $R_{n:m}$ = 1 at n:m = 1:5. In accordance to this explanation, no $R_{n:m}$ bump occurs when data points of the gamma phase time series are made independent by sub-sampling with a period longer than a gamma cycle (*Figure 2—figure supplement 1*), or when extracting phase values from different trials (not shown). As expected, the effect of filtering-induced sinusoidality on $R_{n:m}$ values is stronger for narrower frequency bands (*Figure 2—figure supplement 2*).

Qualitatively similar results were found for 1- and 10 s epochs; however, $R_{n:m}$ values were considerably lower for the latter (*Figure 2B*). In fact, for any fixed n:m ratio and frequency pair, $R_{n:m}$ decreased as a function of epoch length (see *Figure 2C* for $\theta-\gamma_S$ and $R_{1:5}$): the longer the white-noise epoch the more the phase difference distribution becomes uniform. In other words, as standard phase coherence (*Vinck et al., 2010*) and phase-amplitude coupling (*Tort et al., 2010*), phase-phase coupling has positive bias for shorter epochs. As a corollary, notice that false-positive coupling may be detected if control (surrogate) epochs are longer than the original epoch.

## Statistical testing of n:m phase-locking

We next investigated the reliability of surrogate methods for detecting n:m phase-locking (*Figure 2D*). The '*Original*' $R_{n:m}$ value uses the same time window for extracting theta and gamma phases (*Figure 2D*, upper panel). A '*Time Shift*' procedure for creating surrogate epochs has been previously employed (*Belluscio et al., 2012*; *Zheng et al., 2016*), in which the time window for gamma phase is randomly shifted between 1 to 200 ms from the time window for theta phase (*Figure 2D*, upper middle panel). A variant of this procedure is the '*Random Permutation*', in which the time window for gamma phase is randomly chosen (*Figure 2D*, lower middle panel). Finally, in the '*Phase Scramble*' procedure, the timestamps of the gamma phase time series are shuffled (*Figure 2D*, lower panel); clearly, the latter is the least conservative. For each surrogate procedure, $R_{n:m}$ values were obtained by two approaches: '*Single Run*' and '*Pooled*' (*Figure 2E*). In the first approach, each surrogate run (e.g., a time shift or a random selection of time windows) produces one $R_{n:m}$ value (*Figure 2E*, top panels). In the second, $\Delta\varphi_{nm}$ from several surrogate runs are first pooled, then a single $R_{n:m}$ value is computed from the pooled distribution (*Figure 2E*, bottom panel). As illustrated in *Figure 2E*, $R_{n:m}$ computed from a pool of surrogate runs is much smaller than when computed for each individual run. This is due to the dependence of $R_{n:m}$ on the epoch length: pooling instantaneous phase differences across 10 runs of 1 s surrogate epochs is equivalent to analyzing a single surrogate epoch of 10 s. And the longer the analyzed epoch, the more the noise is averaged out and the lower the $R_{n:m}$. Therefore, pooled surrogate epochs summing up to 10 s of total data have lower $R_{n:m}$ than any individual 1 s surrogate epoch.

No phase-phase coupling should be detected in white noise, and therefore *Original* $R_{n:m}$ values should not differ from properly constructed surrogates. However, as shown in *Figure 2F* for $\theta-\gamma_S$ as an illustrative case (similar results hold for any frequency pair), $\theta-\gamma_S$ phase-phase coupling in white noise was statistically significantly larger than in phase-scrambled surrogates (for either *Single Run* or *Pooled* distributions). This was true for surrogate epochs of any length, although the longer the epoch, the lower the actual and the surrogate $R_{n:m}$ values, as expected (compare right and left panels of *Figure 2F*). *Pooled* $R_{1:5}$ distributions derived from either time-shifted (*Figure 2F*) or randomly permutated epochs (not shown) also led to the detection of false positive $\theta-\gamma_S$ phase-phase coupling. On the other hand, *Original* $R_{n:m}$ values were not statistically different from chance

distributions when these were constructed from *Single Run* $R_{n:m}$ values for either *Time Shift* and *Random Permutation* surrogate procedures (*Figure 2F*; see also *Figure 2—figure supplement 3*). We conclude that neither scrambling phases nor pooling individual surrogate epochs should be employed for statistically evaluating n:m phase-locking. Chance distributions should be derived from surrogate epochs of the same length as the original epoch and which preserve phase continuity.

To check if *Single Run* surrogate distributions are capable of statistically detecting true n:m phase-locking, we next simulated noisy Kuramoto oscillators as in *Figure 1*, but of mean natural frequencies set to 8 and 40 Hz. *Original* $R_{1:5}$ values were much greater than the surrogate distribution for coupled – but not uncoupled – oscillators (*Figure 3A*). This result illustrates that variability in the instantaneous frequency leads to low n:m phase-locking levels for independent oscillators even when their mean frequencies are perfect integer multiples. On the other hand, coupled oscillators have high $R_{n:m}$ because variations of their instantaneous frequencies are mutually dependent. We then proceeded to analyze simulated LFPs from a previously published model network (*Kopell et al., 2010*). The network has two inhibitory interneurons, called O and I cells, which spike at theta and gamma frequency, respectively (for a motivation of this model, see *Tort et al., 2007*). Compared to *Single Run* surrogate distributions, the model LFP exhibited significant n:m phase-locking only when the interneurons were coupled; $R_{n:m}$ levels did not differ from the surrogate distribution for the uncoupled network (*Figure 3B*). (Note that the $R_{n:m}$ curve also exhibited a peak for both the uncoupled network and *Single Run* surrogate data, which is due to the low variability in the instantaneous spike frequency of the model cells; without this variability, however, all networks would display perfect n:m phase-locking).

## Spurious n:m phase-locking due to non-sinusoidal waveforms

The simulations above show that *Single Run* surrogates can properly detect n:m phase-locking for oscillators exhibiting variable instantaneous frequency, which is the case of hippocampal theta and gamma oscillations. However, it should be noted that high asymmetry of the theta waveform may also lead to statistically significant $R_{n:m}$ values per se. As illustrated in *Figure 4A*, a non-sinusoidal oscillation such as a theta sawtooth wave can be decomposed into a sum of sine waves at the fundamental and harmonic frequencies, which have decreasing amplitude (i.e., the higher the harmonic frequency, the lower the amplitude). Importantly, the harmonic frequency components are n:m phase-locked to each other: the first harmonic exhibits a fixed 1:2 phase relationship to the fundamental frequency, the second harmonic a 1:3 relationship, and so on (*Figure 4B*). Of note, the higher frequency harmonics not only exhibit cross-frequency phase-phase coupling to the fundamental theta frequency but also phase-amplitude coupling, since they have higher amplitude at the theta phases where the sharp deflection occurs (*Figure 4C* left and *Figure 4—figure supplement 1*; see also *Kramer et al., 2008* and *Tort et al., 2013*).

The gamma-filtered component of a theta sawtooth wave of variable peak frequency thus displays spurious gamma oscillations (i.e., theta harmonics) that have a consistent phase relationship to the theta cycle irrespective of variations in cycle length. In randomly permutated data, however, the theta phases associated with spurious gamma differ from cycle to cycle due to the variability in instantaneous theta frequency. As a result, the spurious n:m phase-coupling induced by sharp signal deflections is significantly higher than the *Random Permutation/Single Run* surrogate distribution (*Figure 4C* right and *Figure 4—figure supplement 2* top row). Interestingly, the significance of this spurious effect is much lower when using the *Time Shift* procedure (*Figure 4—figure supplement 2* bottom row), probably due to the proximity between the original and the time-shifted time series (200 ms maximum distance).

## Assessing n:m phase-locking in actual LFPs

We next proceeded to analyze hippocampal CA1 recordings from seven rats, focusing on the periods of prominent theta activity (active waking and REM sleep). We found similar results between white noise and actual LFP data. Namely, $R_{n:m}$ curves peaked at n:m ratios according to the filtered bands, and $R_{n:m}$ values were lower for longer epochs (*Figure 5A*; compare with *Figure 2B*). As shown in *Figure 5B*, *Original* $R_{n:m}$ values were not statistically different from a proper surrogate distribution (*Random Permutation/Single Run*) in epochs of up to 100 s (but see Figure

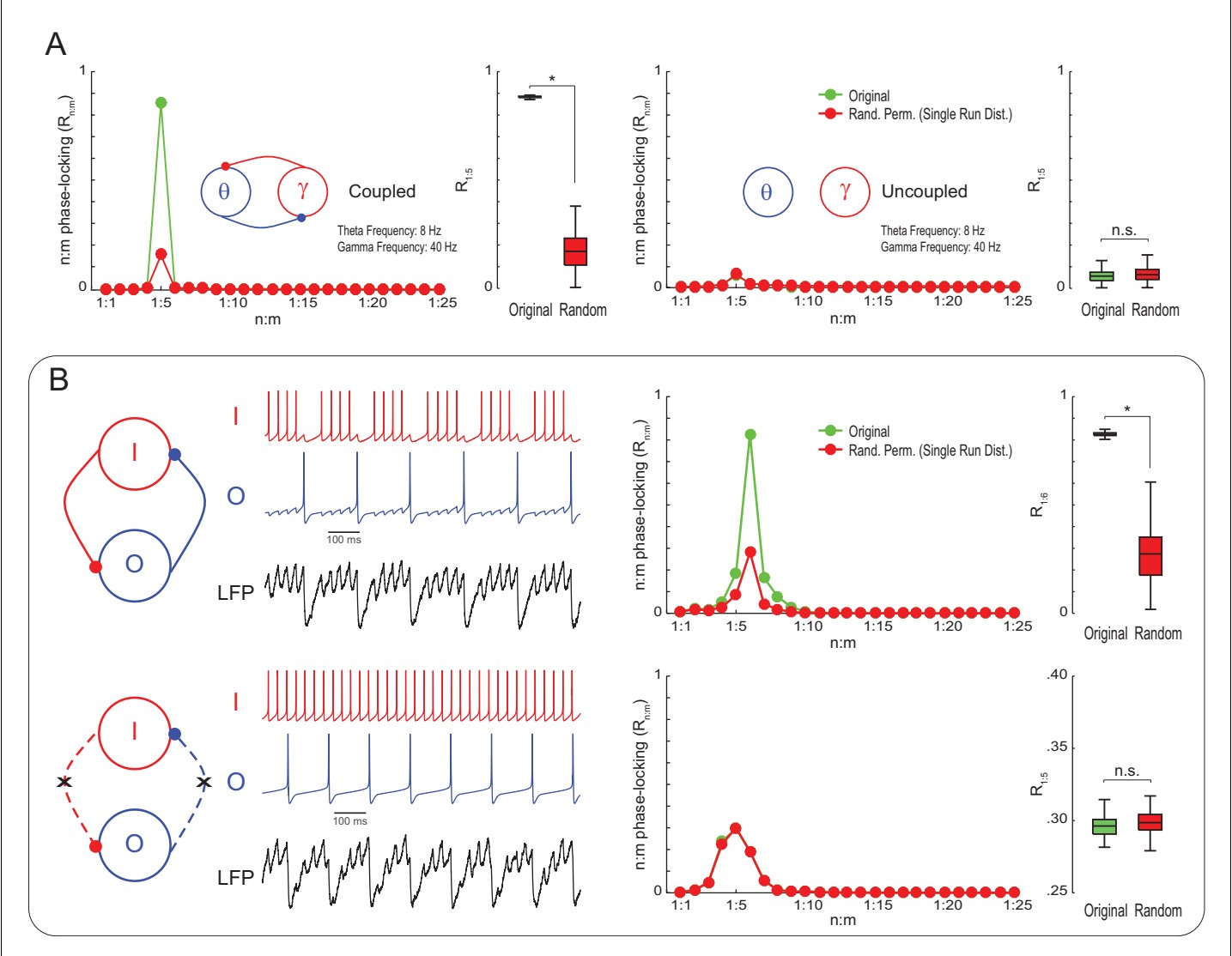

**Figure 3.** True n:m phase-locking leads to significant $R_{n:m}$ values. (**A**) The left panels show mean $R_{n:m}$ curves and distributions of $R_{1:5}$ values for original and surrogate (*Random Permutation/Single Run*) data obtained from the simulation of two coupled Kuramoto oscillators (n = 300; epoch length = 30 s; *p<0.001, t-test). The right panels show the same, but for uncoupled oscillators. In these simulations, each oscillator has instantaneous peak frequency determined by a Gaussian distribution; the mean natural frequencies of the theta and gamma oscillators were set to 8 Hz and 40 Hz, respectively (coupling does not alter the mean frequencies since they already exhibit a 1:5 ratio; compare with *Figure 1*). (**B**) Top panels show results from a simulation of a model network composed of two mutually connected interneurons, O and I cells, which emit spikes at theta and gamma frequency, respectively (*Tort et al., 2007*; *Kopell et al., 2010*). Original n:m phase-locking levels are significantly higher than chance (n = 300; epoch length = 30 s; *p<0.001, t-test). The bottom panels show the same, but for unconnected interneurons. In this case, n:m phase-locking levels are not greater than chance.

10). Noteworthy, as with white-noise data (*Figure 2F*), false positive phase-phase coupling would be inferred if an inadequate surrogate method were employed (*Time Shift/Pooled*) (*Figure 5B*).

We also found no difference between original and surrogate n:m phase-locking levels when employing the metric described in *Sauseng et al. (2009)* (*Figure 5—figure supplement 1*), and when estimating theta phase by interpolating phase values between 4 points of the theta cycle (trough, ascending, peak and descending points) as performed in *Belluscio et al. (2012)* (*Figure 5—figure supplement 2*). The latter was somewhat expected since the phase-phase coupling results in *Belluscio et al. (2012)* did not depend on this particular method of phase estimation (see

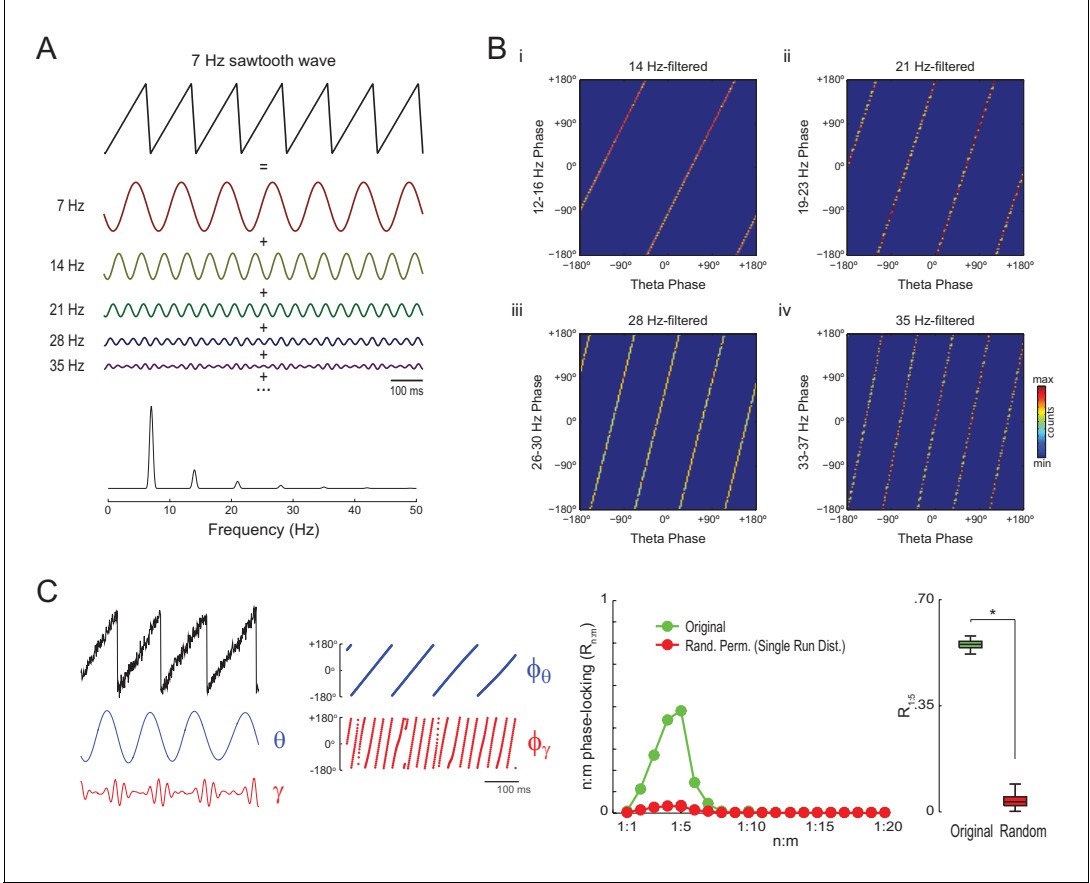

**Figure 4.** Waveform asymmetry may lead to artifactual n:m phase-locking. (**A**) The top traces show a theta sawtooth wave along with its decomposition into a sum of sinusoids at the fundamental (7 Hz) and harmonic (14 Hz, 21 Hz, 28 Hz, 35 Hz, etc) frequencies. The bottom panel shows the power spectrum of the sawtooth wave. Notice power peaks at the fundamental and harmonic frequencies. (**B**) Phase-phase plots (2D histograms of phase counts) for the sawooth wave in A filtered at theta (7 Hz; x-axis phases) and harmonic frequencies (14, 21, 28 and 35 Hz; y-axis phases). (**C**) The left traces show 500 ms of a sawtooth wave along with its theta- and gamma-filtered components and corresponding phase time series. The sawtooth wave was set to have a variable peak frequency, with mean = 8 Hz; no gamma oscillation was added to the signal. Notice that the sharp deflections of the sawtooth wave give rise to artifactual gamma oscillations in the filtered signal (*Kramer et al., 2008*), which have a consistent phase relationship to the theta cycle. The right panels show that artifactual n:m phase-coupling levels induced by the sharp deflections are significantly higher than the chance distribution (n = 300; epoch length = 30 s; *p<0.001, t-test).
The following figure supplements are available for figure 4:

**Figure supplement 1.** Waveform asymmetry may lead to spurious phase-amplitude coupling.
**Figure supplement 2.** The statistical significance of artifactual n:m phase-locking levels induced by waveform asymmetry depends on epoch length and peak frequency variability.

their Figure 6Ce). Moreover, coupling levels did not statistically differ from zero when using the pairwise phase consistency metric described in *Vinck et al. (2010)* (*Figure 5—figure supplement 1*).

We further confirmed our results by analyzing data from three additional rats recorded in an independent laboratory (*Figure 5—figure supplement 3*; see Materials and methods). In addition, we also found similar results in LFPs from other hippocampal layers than *s. pyramidale* (*Figure 5—figure supplement 4*), in neocortical LFPs (not shown), in current-source density (CSD) signals (*Figure 5—figure supplement 4*), in independent components that isolate activity of specific gamma sub-bands (*Schomburg et al., 2014*) (*Figure 5—figure supplement 5*), and in transient gamma bursts (*Figure 5—figure supplement 6*).

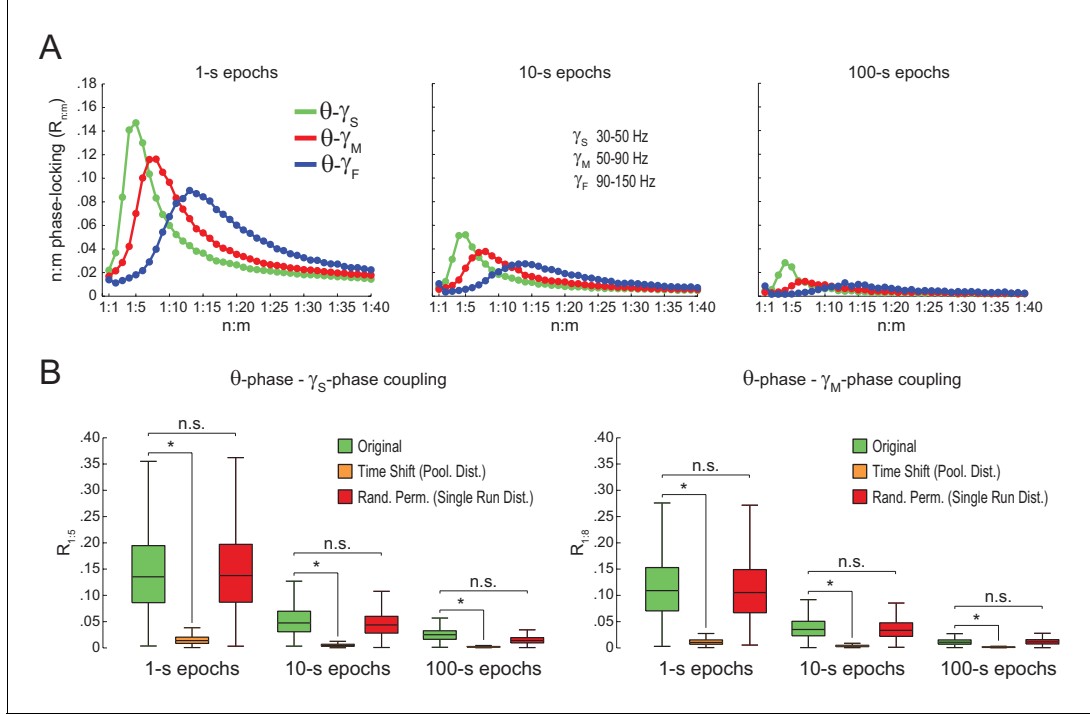

**Figure 5.** Spurious detection of theta-gamma phase-phase coupling in the hippocampus. (**A**) n:m phase-locking levels for actual hippocampal LFPs. Compare with *Figure 2B*. (**B**) Original and surrogate distributions of $R_{n:m}$ values for slow ($R_{1:5}$; left) and middle gamma ($R_{1:8}$; right) for different epoch lengths. The original data is significantly higher than the pooled surrogate distribution, but indistinguishable from the distribution of surrogate values computed using single runs. Similar results hold for fast gamma. *p<0.01, n = 7 animals, Friedman's test with Nemenyi post-hoc test.

The following figure supplements are available for figure 5:

**Figure supplement 1.** Lack of evidence for cross-frequency phase-phase coupling between theta and gamma oscillations using alternative phase-locking metrics.

**Figure supplement 2.** Spurious detection of theta-gamma phase-phase coupling when theta phase is estimated by interpolation.

**Figure supplement 3.** Spurious detection of theta-gamma phase-phase coupling (second dataset).

**Figure supplement 4.** Lack of evidence for theta-gamma phase-phase coupling in all hippocampal layers.

**Figure supplement 5.** Lack of theta-gamma phase-phase coupling in independent components of gamma activity.

**Figure supplement 6.** Lack of theta-gamma phase-phase coupling during transient gamma bursts.

**Figure supplement 7.** The bump in the $R_{n:m}$ curve of hippocampal LFPs highly depends on analyzing contiguous phase time series data.

**Figure supplement 8.** Different filter types give rise to similar results.

## On diagonal stripes in phase-phase plots

Since *Original* $R_{n:m}$ values were not greater than *Single Run* surrogate distributions, we concluded that there is lack of convincing evidence for n:m phase-locking in the hippocampal LFPs analyzed here. However, as in previous reports (*Belluscio et al., 2012*; *Zheng et al., 2016*), phase-phase plots (2D histograms of theta phase vs gamma phase) of actual LFPs displayed diagonal stripes (*Figure 6*), which seem to suggest phase-phase coupling. We next sought to investigate what causes the diagonal stripes in phase-phase plots.

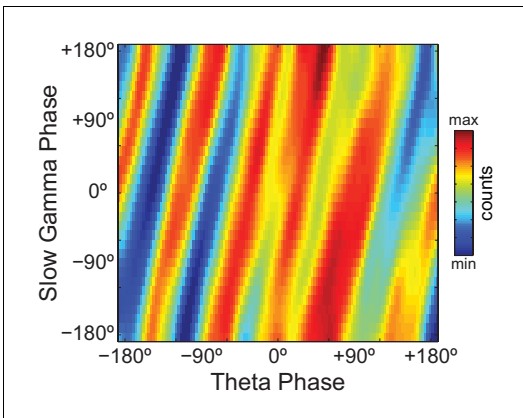

**Figure 6.** Phase-phase plots of hippocampal LFPs display diagonal stripes. Phase-phase plot for theta and slow gamma (average over animals; n = 7 rats). Notice diagonal stripes suggesting phase-phase coupling.

In *Figure 7* we analyze a representative LFP with prominent theta oscillations at ~7 Hz recorded during REM sleep. Due to the non-sinusoidal shape of theta (*Belluscio et al., 2012*; *Sheremet et al., 2016*), the LFP also exhibited spectral peaks at harmonic frequencies (*Figure 7A*). We constructed phase–phase plots using LFP components narrowly filtered at theta and its harmonics: 14, 21, 28 and 35 Hz. Similarly to the sawtooth wave (*Figure 4B*), the phase-phase plots exhibited diagonal stripes whose number was determined by the harmonic order (i.e., the 1st harmonic exhibited two stripes, the second harmonic three stripes, the third, four stripes and the fourth, five stripes; *Figure 7Bi–iv*). Interestingly, when the LFP was filtered at a broad gamma band (30–90 Hz), we observed five diagonal stripes, the same number as when narrowly filtering at 35 Hz; moreover, both gamma and 35 Hz filtered signals exhibited the exact same phase lag (*Figure 7Biv–v*). Therefore, these results indicate that the diagonal stripes in phase-phase plots may be influenced by theta harmonics. Under this interpretation, signals filtered at the gamma band would be likely to exhibit as many stripes as expected for the first theta harmonic

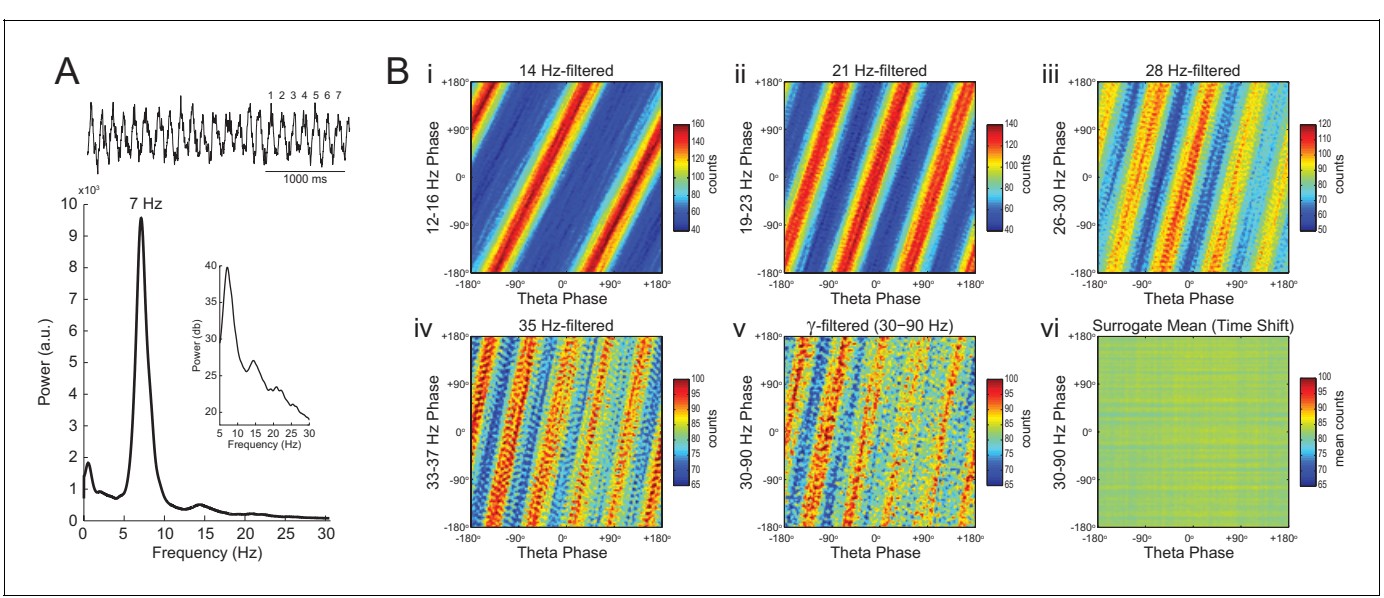

**Figure 7.** Phase–phase coupling between theta and gamma oscillations may be confounded by theta harmonics. (A) Top, representative LFP epoch exhibiting prominent theta activity (~7 Hz) during REM sleep. Bottom, power spectral density. The inset shows power in dB scale. (B) Phase–phase plots for theta and LFP band-pass filtered at harmonic frequencies (14, 21, 28 and 35 Hz), computed using 20 min of concatenated REM sleep. Also shown are phase-phase plots for the conventional gamma band (30–90 Hz) and for the average over individual surrogate runs. Notice that the former mirrors the phase-phase plot of the fourth theta harmonics (35 Hz).

The following figure supplement is available for figure 7:

**Figure supplement 1.** Histogram counts leading to diagonal stripes in phase-phase plots are statistically significant when compared to the distribution of surrogate counts.

falling within the filtered band. Consistent with this possibility, we found that the peak frequency of theta relates to the number of stripes (*Figure 8*).

As in previous studies (*Belluscio et al., 2012*; *Zheng et al., 2016*), phase-phase plots constructed using data averaged from individual time-shifted epochs exhibited no diagonal stripes (*Figure 7Bvi* and *Figure 8*). This is because different time shifts lead to different phase lags; the diagonal stripes of individual surrogate runs that could otherwise be apparent cancel each other out when combining data across multiple runs of different lags (*Figure 8—figure supplement 1*). Moreover, as in *Belluscio et al. (2012)*, the histogram counts that give rise to the diagonal stripes were deemed statistically significant when compared to the mean and standard deviation over individual counts from time-shifted surrogates (*Figure 7—figure supplement 1* and *Figure 8*).

To gain further insight into what generates the diagonal stripes, we next analyzed white-noise signals. As shown in *Figure 9A*, phase-phase plots constructed from filtered white-noise signals also displayed diagonal stripes. Since white noise has no harmonics, these results show that the sinusoidality induced by the filter can by itself lead to diagonal stripes in phase-phase plots, in the same way that it leads to a bump in the $R_{n:m}$ curve (*Figure 2* and *Figure 2—figure supplement 1*). Importantly, as in actual LFPs, bin counts in phase-phase plots of white-noise signals were also deemed statistically significant when compared to the distribution of bin counts from time-shifted surrogates (*Figure 9A*). Since by definition white noise has no n:m phase-locking, we concluded that the statistical analysis of phase-phase plots as originally introduced in *Belluscio et al. (2012)* is too liberal. Nevertheless, we found that phase-phase plots of white noise were no longer statistically significant

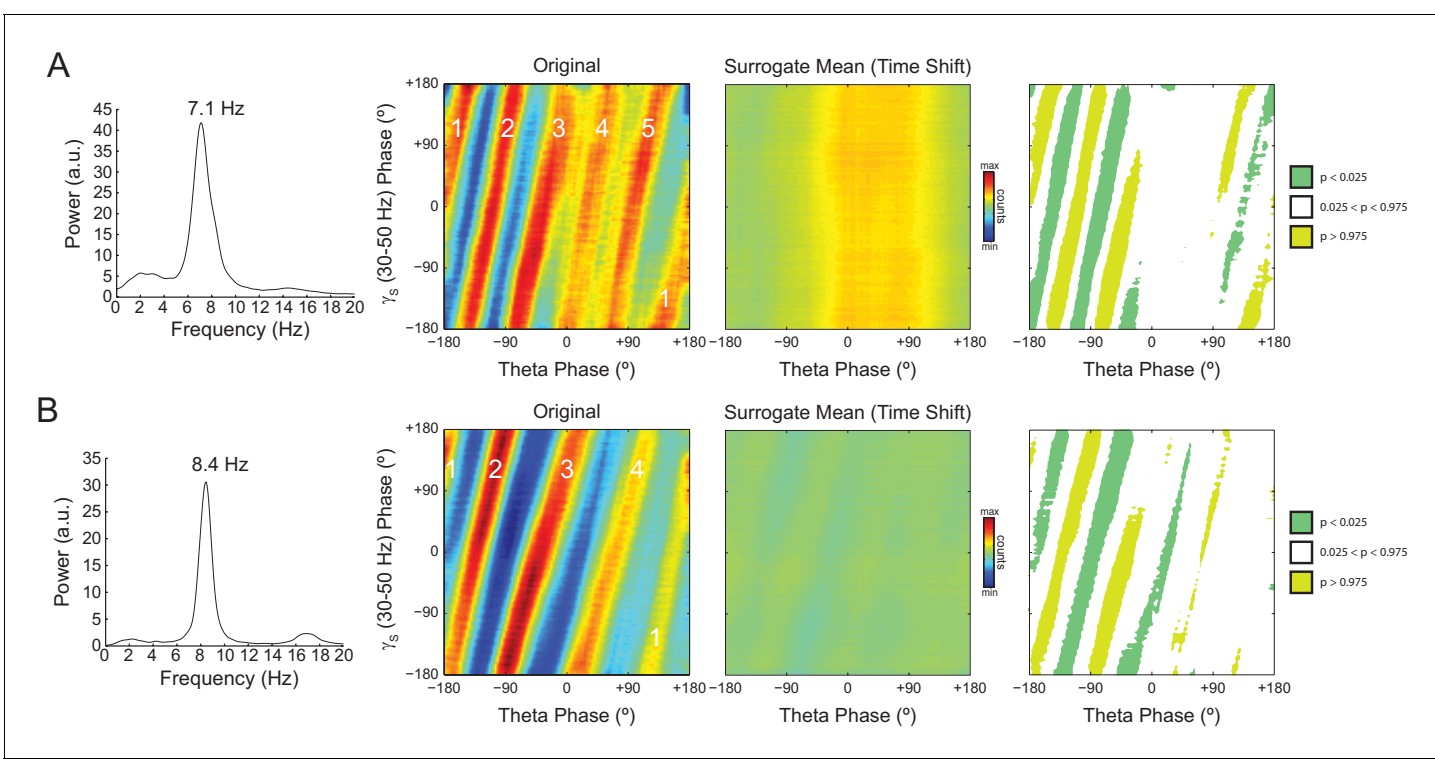

**Figure 8.** The number of stripes in phase-phase plots is determined by the frequency of the first theta harmonic within the filtered gamma range. (**A**) Representative example in which theta has peak frequency of 7.1 Hz. The phase-phase plot between theta and slow gamma (30–50 Hz) exhibits five stripes, since the fourth theta harmonic (35.5 Hz) is the first to fall within 30 and 50 Hz. The rightmost panels show the average phase-phase plot computed over all time-shifted surrogate runs (n = 1000) and the significance of the original plot when compared to the mean and standard deviation over individual surrogate counts, respectively. (**B**) Example in which theta has peak frequency of 8.4 Hz and the phase-phase plot exhibits four stripes, which correspond to the third theta harmonic (33.6 Hz).

The following figure supplement is available for figure 8:

**Figure supplement 1.** Individual time-shifted surrogate runs exhibit diagonal stripes in phase-phase plots.

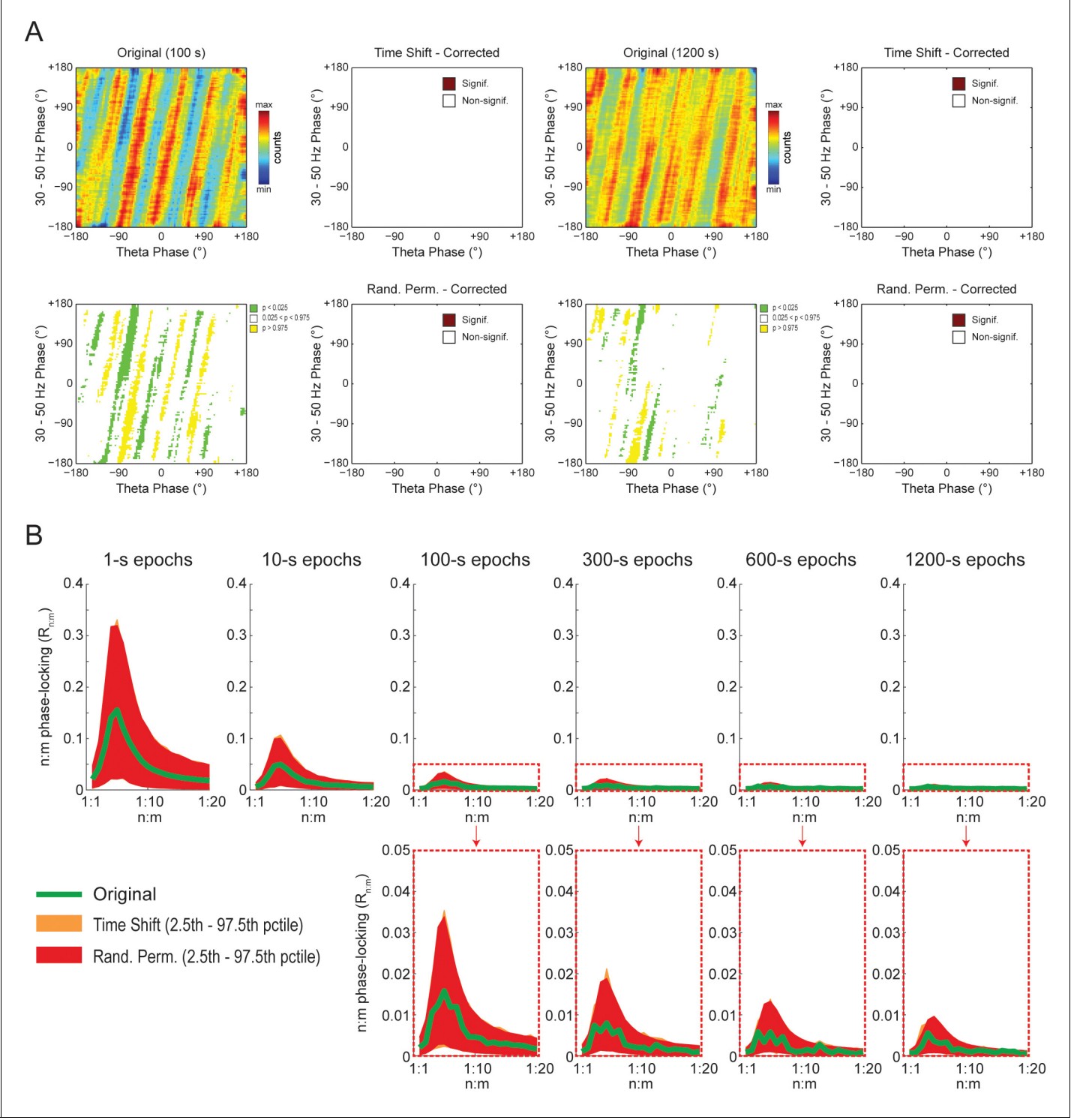

**Figure 9.** Phase-phase plots of white-noise signals display diagonal stripes. (**A**) Representative phase-phase plots computed for white-noise signals. Notice the presence of diagonal stripes for both 100 s (left) and 1200 s (right) epochs. The colormaps underneath show the p-values of the original bin counts when compared to the mean and standard deviation over bin counts of single time-shifted surrogate runs. Also shown are significance maps after correcting for multiple comparisons (Holm-Bonferroni) using either time-shifted (top) or randomly permuted (bottom) surrogate runs. No bin count was considered statistically significant after the correction. (**B**) The top panels show original $R_{n:m}$ curves (green) plotted along with *Single Run* distributions of $R_{n:m}$ curves of time-shifted (orange) and randomly permuted (red) surrogates for different epoch lengths (shades denote the 2.5th–97.5th percentile interval; n = 2100 per distribution). The bottom panels show the same in a zoomed scale.

when using the same approach as in *Belluscio et al. (2012)* but corrected for multiple comparisons (i.e., the number of bins) by the Holm-Bonferroni method (the FDR correction still led to significant bins; not shown). This result was true for different epoch lengths and also when computing surrogate phase-phase plots using the *Random Permutation* procedure (*Figure 9A*). Consistently, for all epoch lengths, *Original* $R_{n:m}$ values fell inside the distribution of *Single Run* surrogate $R_{n:m}$ values computed using either *Time Shift* and *Random Permutation* procedures (*Figure 9B*).

The observations above suggest that the diagonal stripes in phase-phase plots of hippocampal LFPs may actually be caused by filtering-induced sinusoidality, as opposed to being an effect of theta harmonics as we first interpreted. To test this possibility, we next revisited the significance of phase-phase plots of actual LFPs. For epochs of up to 100 s, we found similar results as in white noise, namely, bin counts were no longer statistically significant after correcting for multiple comparisons (Holm-Bonferroni method); this was true when using either the *Time Shift* or *Random Permutation* procedures (*Figure 10A*). Surprisingly, however, when analyzing much longer time series (10 or 20 min of concatenated periods of REM sleep), several bin counts became statistically significant when compared to randomly permutated, but not time-shifted, surrogates (*Figure 10A*). Moreover, this result reflected in the $R_{n:m}$ curves: the *Original* $R_{n:m}$ curve fell within the distribution of *Time Shift/Single Run* surrogate $R_{n:m}$ values for all analyzed lengths, but outside the distribution of *Random Permutation/Single Run* surrogates for the longer time series (*Figure 10B*). We believe such a finding relates to what we observed for synthetic sawtooth waves, in which *Random Permutation* was more sensitive than *Time Shift* to detect the significance of the artifactual coupling caused by waveform asymmetry (*Figure 4—figure supplement 2*). In this sense, the n:m phase-locking between fundamental and harmonic frequencies would persist for small time shifts (±200 ms), albeit in different phase relations, while it would not resist the much larger time shifts obtained through random permutations. However, irrespective of this explanation, it should be noted that since the n:m phase-locking metrics cannot separate artifactual from true coupling, the possibility of the latter cannot be discarded. But if this is the case, we consider unlikely that the very low coupling level (~0.03) would have any physiological significance.

We conclude that the diagonal stripes in phase-phase plots of both white noise and actual LFPs are mainly caused by a temporary n:m alignment of the phase time-series secondary to the filtering-induced sinusoidality, and as such they are also apparent in surrogate data (*Figure 8—figure supplement 1* and *Figure 10—figure supplement 1*). However, for actual LFPs there is a second influence, which can only be detected when analyzing very long epoch lengths, and which we believe is due to theta harmonics.

## Discussion

Theta and gamma oscillations are hallmarks of hippocampal activity during active exploration and REM sleep (*Buzsáki et al., 2003*; *Csicsvari et al., 2003*; *Montgomery et al., 2008*). Theta and gamma are well known to interact by means of phase-amplitude coupling, in which the instantaneous gamma amplitude waxes and wanes as a function of theta phase (*Bragin et al., 1995*; *Scheffer-Teixeira et al., 2012*; *Caixeta et al., 2013*). This particular type of cross-frequency coupling has been receiving large attention and related to functional roles (*Canolty and Knight, 2010*; *Hyafil et al., 2015*). In addition to phase-amplitude coupling, theta and gamma oscillations can potentially interact in many other ways (*Jensen and Colgin, 2007*; *Hyafil et al., 2015*). For example, the power of slow gamma oscillations may be inversely related to theta power (*Tort et al., 2008*), suggesting amplitude-amplitude coupling. Recently, it has been reported that theta and gamma in hippocampal LFPs would also couple by means of n:m phase-locking (*Belluscio et al., 2012*; *Zheng and Zhang, 2013*; *Xu et al., 2013*, *2015*; *Zheng et al., 2016*). Among other implications, this finding was taken as evidence for network models of working memory (*Lisman and Idiart, 1995*; *Jensen and Lisman, 2005*; *Lisman, 2005*) and for a role of basket cells in generating cross-frequency coupling (*Belluscio et al., 2012*; *Buzsáki and Wang, 2012*). However, our results show a lack of convincing evidence for n:m phase-locking in the two hippocampal datasets analyzed here, and further suggest that previous work may have spuriously detected phase-phase coupling due to an improper use of surrogate methods, a concern also raised for phase-amplitude coupling (*Aru et al., 2015*).

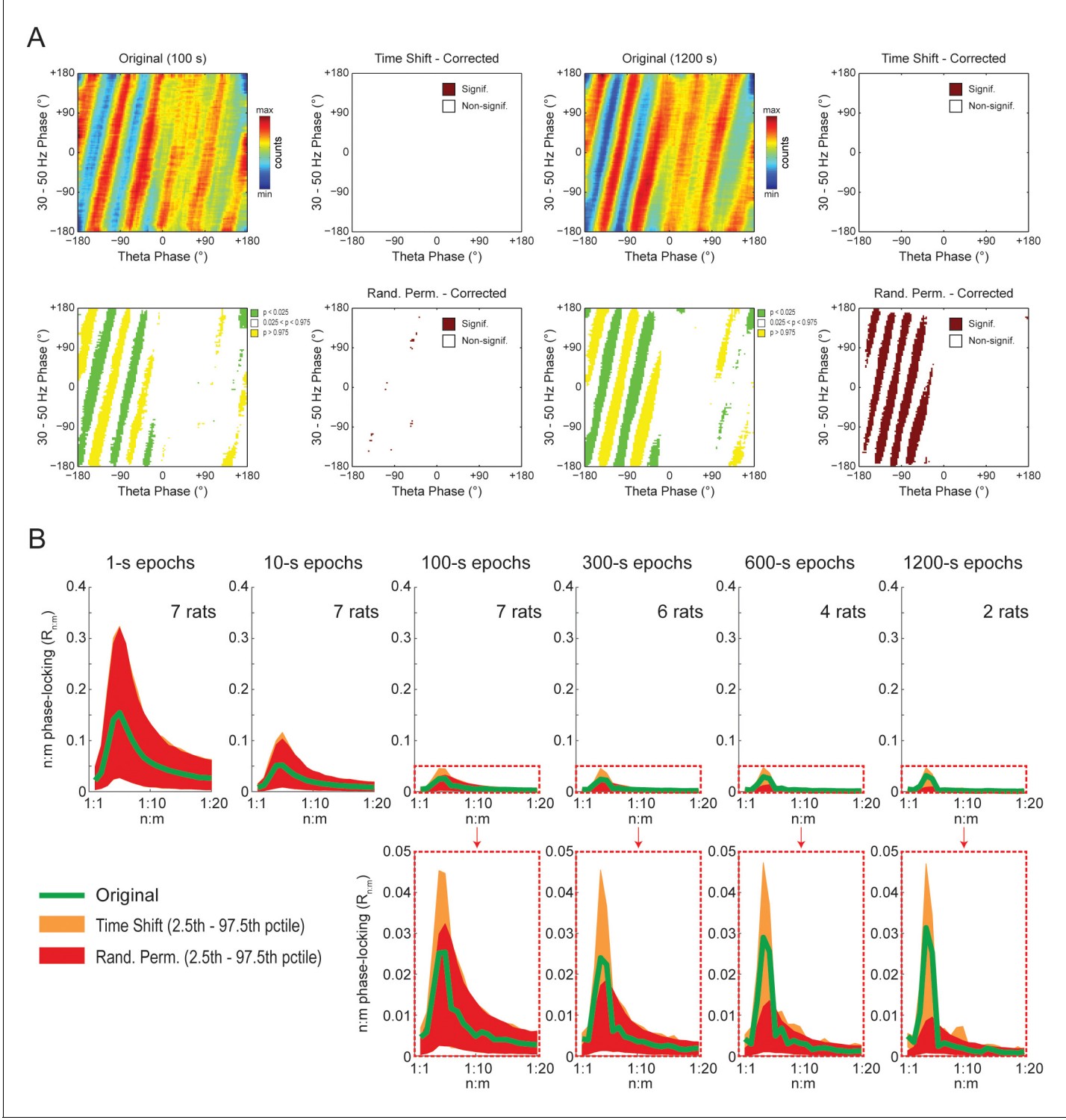

**Figure 10.** Weak but statistically significant n:m phase-locking can be detected when analyzing long LFP epochs (>100 s). (**A**) Panels show the same as in **Figure 9A** but for a representative hippocampal LFP. Notice that several bin counts of the 1200 s epoch remain statistically significant after correction for multiple comparisons (Holm-Bonferroni) when compared to randomly permutated, but not time-shifted, surrogates (bottom right plot). (**B**) As in **Figure 9B**, but for actual LFPs (n = 300 samples per animal; the number of analyzed animals is stated in each panel). For the very long epochs, notice that the original $R_{n:m}$ curve falls within the distribution of time-shifted surrogates but outside the distribution of randomly permutated ones.

*Figure 10 continued on next page*

*Figure 10 continued*

The following figure supplements are available for figure 10:

**Figure supplement 1.** *Random Permutation* leads to less visible diagonal stripes than *Time Shift* in phase-phase plots of long LFP epochs.

**Figure supplement 2.** Diagonal stripes in phase-phase plots depend on analyzing contiguous phase time series data.

## Statistical inference of phase-phase coupling

When searching for phase-phase coupling between theta and gamma, we noticed that our $R_{n:m}$ values differed from those reported in previous studies (*Belluscio et al., 2012*; *Xu et al., 2013*, *2015*; *Zheng et al., 2016*). We suspected that this could be due to differences in the duration of the analyzed epochs. We then investigated the dependence of $R_{n:m}$ on epoch length, and found a strong positive bias for shorter epochs. In addition, $R_{n:m}$ values exhibit greater variability across samples as epoch length decreases for both white noise and actual data (e.g., compare in *Figure 5B* the data dispersion in *Original* $R_{1:5}$ or $R_{1:8}$ boxplots for different epoch lengths). Since theta and gamma peak frequencies are not constant in these signals, the longer the epoch, the more the theta and gamma peak frequencies are allowed to fluctuate and the more apparent the lack of coupling. On the other hand, $\Delta\varphi_{nm}$ distribution becomes less uniform for shorter epochs. The dependence of n:m phase-coupling metrics on epoch length has important implications in designing surrogate epochs for testing the statistical significance of actual $R_{n:m}$ values. Of note, methodological studies on 1:1 phase-synchrony have properly used single surrogate runs of the same length as the original signal (*Le Van Quyen et al., 2001*; *Hurtado et al., 2004*). As demonstrated here, spurious detection of phase-phase coupling may occur if surrogate epochs are longer than the original epoch. This is the case when one lumps together several surrogate epochs before computing $R_{n:m}$. When employing proper controls, our results show that $R_{n:m}$ values of real data do not differ from surrogate values in theta epochs of up to 100 s. Moreover, the prominent bump in the $R_{n:m}$ curve disappears when subsampling data at a lower frequency than gamma for both white noise and hippocampal LFPs (see *Figure 2—figure supplement 1* and *Figure 5—figure supplement 7*), which suggests that it is due to the statistical dependence among contiguous data points introduced by the filter (which we referred to as 'filtering-induced sinusoidality').

Therefore, even though the n:m phase-locking metric $R_{n:m}$ is theoretically well-defined and varies between 0 and 1, an estimated $R_{n:m}$ value in isolation does not inform if two oscillations exhibit true phase-coupling or not. This can only be inferred after testing the statistical significance of the estimated $R_{n:m}$ value against a proper surrogate distribution (but notice that false-positive cases may occur due to waveform asymmetry; *Figure 4C*). While constructing surrogate data renders the metric computationally more expensive, such an issue is not specific for measuring n:m phase-locking but also happens for other metrics commonly used in the analysis of neurophysiological data, such as coherence, spike-field coupling, phase-amplitude coupling, mutual information and directionality measures, among many others (*Le Van Quyen et al., 2001*; *Hurtado et al., 2004*; *Pereda et al., 2005*; *Tort et al., 2010*).

The recent studies assessing theta-gamma phase-phase coupling in hippocampal LFPs have not tested the significance of individual $R_{n:m}$ values against chance (*Belluscio et al., 2012*; *Zheng and Zhang, 2013*; *Xu et al., 2013*, *2015*; *Zheng et al., 2016*). Two studies (*Belluscio et al., 2012*; *Zheng et al., 2016*) statistically inferred the existence of n:m phase-locking by comparing empirical phase-phase plots with those obtained from the average of 1000 time-shifted surrogate runs. Specifically, *Belluscio et al. (2012)* established a significance threshold for each phase-phase bin based on the mean and standard deviation of individual surrogate counts in that bin, and showed that the bin counts leading to diagonal stripes were statistically significant. Here we were able to replicate these results (*Figure 7—figure supplement 1* and *Figure 8*). However, we note that a phase-phase bin count is not a metric of n:m phase-locking; it does not inform coupling strength and even coupled oscillators have bins with non-significant counts. A bin count would be analogous to a phase difference vector ($e^{i\Delta\varphi_{nm}(t)}$), which is also not a metric of n:m phase-locking per se, but used to compute one. That is, in the same way that the $R_{n:m}$ considers all phase difference vectors, n:m phase-locking can only be inferred when considering all bin counts in a phase-phase plot. In this sense, by

analyzing the phase-phase plot as a whole, it was assumed that the appearance diagonal stripes was due to theta-gamma coupling; no such stripes were apparent in phase-phase plots constructed from the average over all surrogate runs (see Figure 6A in *Belluscio et al., 2012*). However, here we showed that single time-shifted surrogate runs do exhibit diagonal stripes (*Figure 8—figure supplement 1* and *Figure 10—figure supplement 1*), that is, similar stripes exist at the level of a *Single Run* surrogate analysis, in the same way that *Single Run* surrogates also exhibit a bump in the $R_{n:m}$ curve. Averaging 1000 surrogate phase-phase plots destroys the diagonal stripes since different time shifts lead to different phase lags. Moreover, since the average is the sum divided by a scaling factor (the sample size), computing the average phase-phase plot is equivalent to computing a single phase-phase plot using the pool of all surrogate runs, which is akin to the issue of computing a single $R_{n:m}$ value from a pooled surrogate distribution (*Figure 2*). Note that even bin counts in phase-phase plots of white noise are considered significant under the statistical analysis introduced in *Belluscio et al. (2012)* (*Figure 9A*). Nevertheless, this was no longer the case when adapting their original framework to include a Holm-Bonferroni correction for multiple comparisons (*Figure 9A*).

Here we showed that the presence of diagonal stripes in phase-phase plots is not sufficient to conclude the existence of phase-phase coupling. The diagonal stripes are simply a visual manifestation of a maintained phase relationship, and as such they essentially reflect what $R_{n:m}$ measures: that is, the 'clearer' the stripes, the higher the $R_{n:m}$. Therefore, in addition to true coupling, the same confounding factors that influence $R_{n:m}$ also influence phase-phase plots, such as filtering-induced sinusoidality and frequency harmonics. Our results suggest that the former is a main factor, because white-noise signals have no harmonics but nevertheless display stripes in phase-phase plots (*Figure 9A*). In accordance, no stripes are observed in phase-phase plots of white noise when subsampling the time series (*Figure 10—figure supplement 2*; see also *Figure 2—figure supplement 1*). However, in actual LFPs filtering is not the only influence: (1) for the same filtered gamma band (30–50 Hz), the number of stripes relates to theta frequency (*Figure 8*); (2) for very long time series (i.e., 10–20 min of concatenated data), the stripes in phase-phase plots of actual data – but not of white noise – persist after correcting for multiple comparisons when employing *Random Permutation/Single Run* surrogates (*Figure 10A*); (3) a striped-like pattern remains in phase-phase plots of actual LFPs after subsampling the time series (*Figure 10—figure supplement 2*). Consistently, $R_{n:m}$ values of actual LFPs are greater than those of white noise in 1200 s epochs (~0.03 vs ~0.005, compare the bottom right panels of *Figures 9B* and *10B*). Interestingly, *Original* $R_{n:m}$ values of actual LFPs are not statistically different from the distribution of *Time Shift/Single Run* surrogates even for the very long epochs (*Figure 10B*), which suggests that *Random Permutation* is more powerful than *Time Shift* and should therefore be preferred. Though a very weak but true coupling effect cannot be discarded, based on our analysis of sawtooth waves (*Figure 4* and *Figure 4—figure supplement 2*), we believe these results can be explained by theta harmonics, which would remain phase-locked to the fundamental frequency under small time shifts. Sharp signal deflections have been previously recognized to generate artifactual phase-amplitude coupling (*Kramer et al., 2008*; *Scheffer-Teixeira et al., 2013*; *Tort et al., 2013*; *Aru et al., 2015*; *Lozano-Soldevilla et al., 2016*). Interestingly, *Hyafil (2015)* recently suggested that the non-sinusoidality of alpha waves could underlie the 1:2 phase-locking between alpha and beta observed in human EEG (*Nikulin and Brismar, 2006*; see also *Palva et al., 2005*). To the best of our knowledge, there is currently no metric capable of automatically distinguishing true cross-frequency coupling from waveform-induced artifacts in collective signals such as LFP, EEG and MEG signals. Ideally, learning how the signal is generated from the activity of different neuronal populations would answer whether true cross-frequency coupling exists or not (*Hyafil et al., 2015*), but unfortunately this is methodologically challenging.

## Lack of evidence vs evidence of non-existence

One could argue that we did not analyze a proper dataset, or else that prominent phase-phase coupling would only occur during certain behavioral states not investigated here. We disagree with these arguments for the following reasons: (1) we could reproduce our results using a second dataset from an independent laboratory (*Figure 5—figure supplement 3*), and (2) we examined the same behavioral states in which n:m phase-locking was reported to occur (active waking and REM sleep). One could also argue that there exists multiple gammas, and that different gamma types are most prominent in different hippocampal layers (*Colgin et al., 2009*; *Scheffer-Teixeira et al., 2012*; *Tort et al., 2013*; *Schomburg et al., 2014*; *Lasztóczi and Klausberger, 2014*); therefore, prominent

theta-gamma phase-phase coupling could exist in other hippocampal layers not investigated here. We also disagree with this possibility because: (1) we examined the same hippocampal layer in which theta-gamma phase-phase coupling was reported to occur (*Belluscio et al., 2012*); moreover, (2) we found similar results in all hippocampal layers (we recorded LFPs using 16-channel silicon probes, see Materials and methods) (*Figure 5—figure supplement 4*) and (3) in parietal and entorhinal cortex recordings (not shown). Furthermore, similar results hold when (4) filtering LFPs within any gamma sub-band (*Figure 5* and *Figure 5—figure supplement 1* to *6*), (5) analyzing CSD signals (*Figure 5—figure supplement 4*), or (6) analyzing independent components that maximize activity within particular gamma sub-bands (*Schomburg et al., 2014*) (*Figure 5—figure supplement 5*). Finally, one could argue that gamma oscillations are not continuous but transient, and that assessing phase-phase coupling between theta and transient gamma bursts would require a different type of analysis than employed here. Regarding this argument, we once again stress that we used the exact same methodology as originally used to detect theta-gamma phase-phase coupling (*Belluscio et al., 2012*). Nevertheless, we also ran analysis only taking into account periods in which gamma amplitude was >2 SD above the mean ('gamma bursts') and found no statistically significant phase-phase coupling (*Figure 5—figure supplement 6*).

Following *Belluscio et al. (2012)*, other studies also reported theta-gamma phase-phase coupling in the rodent hippocampus (*Zheng and Zhang, 2013*; *Xu et al., 2013*, *2015*; *Zheng et al., 2016*) and amygdala (*Stujenske et al., 2014*). In addition, human studies had previously reported theta-gamma phase-phase coupling in scalp EEG (*Sauseng et al., 2008*, *2009*; *Holz et al., 2010*). Most of these studies, however, have not tested the statistical significance of coupling levels against chance (*Sauseng et al., 2008*, *2009*; *Holz et al., 2010*; *Zheng and Zhang, 2013*; *Xu et al., 2013*, *2015*; *Stujenske et al., 2014*), while *Zheng et al. (2016)* based their statistical inferences on the inspection of diagonal stripes in phase-phase plots as originally introduced in *Belluscio et al. (2012)*. We further note that epoch length was often not informed in the animal studies. Based on our results, we believe that differences in analyzed epoch length are likely to explain the high variability of $R_{n:m}$ values across different studies, from ~0.4 (*Zheng et al., 2016*) down to 0.02 (*Xu et al., 2013*).

Since it is philosophically impossible to prove the absence of an effect, the burden of proof should be placed on demonstrating that a true effect exists. In this sense, and to the best of our knowledge, none of previous research investigating theta-gamma phase-phase coupling has properly tested $R_{n:m}$ against chance. Many studies have focused on comparing changes in n:m phase-locking levels, but we believe these can be influenced by other variables such as changes in power, which affect the signal-to-noise ratio and consequently also the estimation of the phase time series. Interestingly, in their pioneer work, Tass and colleagues used filtered white noise to construct surrogate distributions and did not find significant n:m phase-locking among brain oscillations (*Tass et al., 1998*, *2003*). On the other hand, it is theoretically possible that n:m phase-locking exists but can only be detected by other types of metrics yet to be devised. In any case, our work shows that there is currently no convincing evidence for genuine theta-gamma phase-phase coupling using the same phase-locking metric ($R_{n:m}$) as employed in previous studies (*Belluscio et al., 2012*; *Zheng and Zhang, 2013*; *Xu et al., 2013*, *2015*; *Stujenske et al., 2014*; *Zheng et al., 2016*), at least when examining LFP epochs of up to 100 s of prominent theta activity. For longer epoch lengths, though, we did find that $R_{n:m}$ values of hippocampal LFPs may actually differ from those of randomly permuted, but not time-shifted, surrogates (*Figure 10B*). While we tend to ascribe such result to the effect of theta harmonics, we note that the possibility of true coupling cannot be discarded. But we are particularly skeptical that the very low levels of coupling strength observed in long LFP epochs would be physiologically meaningful.

## Implications for models of neural coding by theta-gamma coupling

*Lisman and Idiart (1995)* proposed an influential model in which theta and gamma oscillations would interact to produce a neural code. The theta-gamma coding model has since been improved (*Jensen and Lisman, 2005*; *Lisman, 2005*; *Lisman and Buzsáki, 2008*), but its essence remains the same (*Lisman and Jensen, 2013*): nested gamma cycles would constitute memory slots, which are parsed at each theta cycle. Accordingly, *Lisman and Idiart (1995)* hypothesized that working memory capacity (7 ± 2) is determined by the number of gamma cycles per theta cycle.

Both phase-amplitude and phase-phase coupling between theta and gamma have been considered experimental evidence for such coding scheme (*Lisman and Buzsáki, 2008*; *Sauseng et al., 2009*; *Axmacher et al., 2010*; *Belluscio et al., 2012*; *Lisman and Jensen, 2013*; *Hyafil et al., 2015*; *Rajji et al., 2016*). In the case of phase-amplitude coupling, the modulation of gamma amplitude within theta cycles would instruct a reader network when the string of items represented in different gamma cycles starts and terminates. On the other hand, the precise ordering of gamma cycles within theta cycles that is consistent across theta cycles would imply phase-phase coupling; indeed, n:m phase-locking is a main feature of computational models of sequence coding by theta-gamma coupling (*Lisman and Idiart, 1995*; *Jensen and Lisman, 1996*; *Jensen et al., 1996*). In contrast to these models, however, our results show that the theta phases in which gamma cycles begin/end are not fixed across theta cycles, which is to say that gamma cycles are not precisely timed but rather drift; in other words, gamma is not a clock (*Burns et al., 2011*).

If theta-gamma neural coding exists, our results suggest that the precise location of gamma memory slots within a theta cycle is not required for such a code, and that the ordering of the represented items would be more important than the exact spike timing of the cell assemblies that represent the items (*Lisman and Jensen, 2013*).

## Conclusion

In summary, while absence of evidence is not evidence of absence, our results challenge the hypothesis that theta-gamma phase-phase coupling exists in the hippocampus. At best, we only found significant $R_{n:m}$ values when examining long LFP epochs (>100 s), but these had very low magnitude (and we particularly attribute their statistical significance to the effects of harmonics). We believe that the evidence in favor of n:m phase-locking in other brain regions and signals could potentially also be explained by simpler effects (e.g., filtering-induced sinusoidality, asymmetrical waveform, and improper statistical tests). While no current technique can differentiate spurious from true phase-phase coupling, previous findings should be revisited and, whenever suitable, checked against the confounding factors and the more conservative surrogate procedures outlined here.

# Materials and methods

## Animals and surgery

All procedures were approved by our local institutional ethics committee (Comissão de Ética no Uso de Animais - CEUA/UFRN, protocol number 060/2011) and were in accordance with the National Institutes of Health guidelines. We used seven male Wistar rats (2–3 months; 300–400 g) from our breeding colony, kept under 12 hr/12 hr dark-light cycle. We recorded from the dorsal hippocampus through either multi-site linear probes (n = 6 animals; 4 probes had 16 4320 $\mu m^2$ contacts spaced by 100 $\mu m$; 1 probe had 16 703 $\mu m^2$ contacts spaced by 100 $\mu m$; 1 probe had 16 177 $\mu m^2$ contacts spaced by 50 $\mu m$; all probes from NeuroNexus) or single wires (n = 1 animal; 50 $\mu m$ diameter) inserted at AP −3.6 mm and ML 2.5 mm. Results shown in the main figures were obtained for LFP recordings from the CA1 pyramidal cell layer, identified by depth coordinate and characteristic electrophysiological benchmarks such as highest ripple power (see *Figure 5—figure supplement 4* for an example). Similar results were obtained for recordings from other hippocampal layers (*Figure 5—figure supplement 4*).

We also analyzed data from three additional rats downloaded from the Collaborative Research in Computational Neuroscience data sharing website (www.crcns.org) (*Figure 5—figure supplement 3*). These recordings are a generous contribution by György Buzsáki's laboratory (HC3 dataset, *Mizuseki et al., 2013*, *2014*).

## Data collection

Recording sessions were performed in an open field (1 m x 1 m) and lasted 4–5 hr. Raw signals were amplified (200x), filtered between 1 Hz and 7.5 kHz (third order Butterworth filter), and digitized at 25 kHz (RHA2116, IntanTech). The LFP was obtained by further filtering between 1–500 Hz and downsampling to 1000 Hz.

## Data analysis

Active waking and REM sleep periods were identified from spectral content (high theta/delta power ratio) and video recordings (movements during active waking; clear sleep posture and preceding slow-wave sleep for REM). The results were identical for active waking and REM epochs; throughout this work we only show the latter. The analyzed REM sleep dataset is available at 10.5061/dryad. 12t21. MATLAB codes for reproducing our analyses are available at https://github.com/tortlab/phase_phase .

We used built-in and custom-written MATLAB routines. Band-pass filtering was obtained using a least squares finite impulse response (FIR) filter by means of the 'eegfilt' function from the EEGLAB Toolbox (*Delorme and Makeig, 2004*). The filter order was three times the sampling rate divided by the low cutoff frequency. The eegfilt function calls the MATLAB 'filtfilt' function, which applies the filter forward and then again backwards to ensure no distortion of phase values. Similar results were obtained when employing other types of filters (*Figure 5—figure supplement 8*).

The phase time series was estimated through the Hilbert transform. To estimate the instantaneous theta phase of actual data, we filtered the LFP between 4–20 Hz, a bandwidth large enough to capture theta wave asymmetry (*Belluscio et al., 2012*). Estimating theta phase by the interpolation method described in *Belluscio et al. (2012)* led to similar results (*Figure 5—figure supplement 2*).

The CSD signals analyzed in *Figure 5—figure supplement 4* were obtained as −A +2B −C, where A, B and C denote LFP signals recorded from adjacent probe sites. In *Figure 5—figure supplement 5*, the independent components were obtained as described in *Schomburg et al. (2014)*; phase-amplitude comodulograms were computed as described in *Tort et al. (2010)*.

## n:m phase-locking

We measured the consistency of the phase difference between accelerated time series $(\Delta\varphi_{nm}(t_j) = n*\varphi_\gamma(t_j) - m*\varphi_\theta(t_j))$. To that end, we created unitary vectors whose angle is the instantaneous phase difference $(e^{i\Delta\varphi_{nm}(t_j)})$, where $j$ indexes the time sample, and then computed the length of the mean vector: $R_{n:m} = \left\|\frac{1}{N}\sum_{j=1}^{N} e^{i\Delta\varphi_{nm}(t_j)}\right\|$, where N is the total number of time samples (epoch length in seconds x sampling frequency in Hz). $R_{n:m}$ equals 1 when $\Delta\varphi_{nm}$ is constant for all time samples $t_j$, and 0 when $\Delta\varphi_{nm}$ is uniformly distributed. This metric is also commonly referred to as 'mean resultant length' or 'mean radial distance' (*Belluscio et al., 2012*; *Stujenske et al., 2014*; *Zheng et al., 2016*). Qualitatively similar results were obtained when employing the framework introduced in *Sauseng et al. (2009)*, which computes the mean radial distance using gamma phases in separated theta phase bins, or the pairwise phase consistency metric described in *Vinck et al. (2010)* (*Figure 5—figure supplement 1*). Phase-phase plots were obtained by first binning theta and gamma phases into 120 bins and next constructing 2D histograms of phase counts, which were smoothed using a Gaussian kernel of σ = 10 bins.

## Surrogates

In all cases, theta phase was kept intact while gamma phase was mocked in three different ways: (1) *Time Shift:* the gamma phase time series is randomly shifted between 1 and 200 ms; (2) *Random Permutation:* a contiguous gamma phase time series of the same length as the original is randomly extracted from the same session. (3) *Phase Scrambling*: the timestamps of the gamma phase time series are randomly shuffled (thus not preserving phase continuity). For each case, $R_{n:m}$ values were computed using either $\Delta\varphi_{nm}$ distribution for single surrogate runs (*Single Run Distribution*) or the pooled distribution of $\Delta\varphi_{nm}$ over 100 surrogate runs (*Pooled Distribution*).

For each animal, behavioral state (active waking or REM sleep) and epoch length, we computed 300 *Original* $R_{n:m}$ values using different time windows along with 300 mock $R_{n:m}$ values per surrogate method. Therefore, in all figures each boxplot was constructed using the same number of samples (=300 x number of animals). For instance, in *Figure 5B* we used n = 7 animals x 300 samples per animal = 2100 samples (but see *Statistics* below). In *Figure 2*, boxplot distributions for the white-noise data were constructed using n = 2100.

## Simulations

Kuramoto oscillators displaying n:m phase-locking were modeled as described in *Osipov et al. (2007)*:

$$\dot{\varphi}_\theta = \omega_\theta + \varepsilon \sin\left(n\varphi_\gamma - m\varphi_\theta\right)$$
$$\dot{\varphi}_\gamma = \omega_\gamma + \varepsilon \sin\left(m\varphi_\theta - n\varphi_\gamma\right),$$

where $\varepsilon$ is the coupling strength and $\omega_\theta$ and $\omega_\gamma$ are the natural frequencies of theta and gamma, respectively, which followed a Gaussian probability ($\sigma$ = 5 Hz) at each time step. We used $\varepsilon$ = 10, n = 1, m = 5, and dt = 0.001 s. The mean theta and gamma frequencies of each simulation are stated in the main text. For uncoupled oscillators, we set $\varepsilon$ = 0.

For implementing the O-I cell network (*Figure 3B*), we simulated the model previously described in *Kopell et al. (2010)*. We used the same parameters as in Figure 3A of *Kopell et al. (2010)*, with white noise ($\sigma$ = 0.001) added to the I cell drive to create variations in spike frequency. NEURON (https://www.neuron.yale.edu/) codes for the model are available at ModelDB (https://senselab.med.yale.edu/).

The sawtooth wave in *Figure 4C* was simulated using dt = 0.001 s. Its instantaneous frequency followed a Gaussian distribution with mean = 8 Hz and $\sigma$ = 5 Hz; white noise ($\sigma$ = 0.1) was added to the signal.

In *Figures 3* and *4C*, boxplot distributions for simulated data were constructed using n = 300.

## Statistics

For white noise data (*Figure 2F*), given the large sample size (n = 2100) and independence among samples, we used one-way ANOVA with Bonferroni post-hoc test. For statistical analysis of real data (*Figure 5B*), we avoided nested design and inflation of power and used the mean $R_{n:m}$ value per animal. In this case, due to the reduced sample size (n = 7) and lack of evidence of normal distribution (Shapiro-Wilk normality test), we used the Friedman's test and Nemenyi post-hoc test. In *Figures 3* and *4C*, we tested if $R_{n:m}$ values of simulated data were greater than the distribution of surrogate values using one-tailed t-tests.

## Acknowledgements

The authors are indebted to the reviewers for many constructive comments and helpful suggestions. The authors are grateful to Jurij Brankačk and Andreas Draguhn for donation of NeuroNexus probes. Supported by Conselho Nacional de Desenvolvimento Científico e Tecnológico (CNPq) and Coordenação de Aperfeiçoamento de Pessoal de Nível Superior (CAPES). The authors declare no competing financial interests.

## Additional information

### Funding

| Funder | Author |
| --- | --- |
| Conselho Nacional de Desenvolvimento Científico e Tecnológico | Robson Scheffer-Teixeira<br>Adriano BL Tort |
| Coordenação de Aperfeiçoamento de Pessoal de Nível Superior | Robson Scheffer-Teixeira<br>Adriano BL Tort |

The funders had no role in study design, data collection and interpretation, or the decision to submit the work for publication.

### Author contributions

RS-T, Acquisition of data, Analysis and interpretation of data, Drafting or revising the article; ABLT, Conception and design, Analysis and interpretation of data, Drafting or revising the article

## Author ORCIDs

Adriano BL Tort, http://orcid.org/0000-0002-9877-7816

## Ethics

Animal experimentation: All procedures were approved by the institutional ethics committee of Federal University of Rio Grande do Norte (Comissão de Ética no Uso de Animais - CEUA/UFRN, protocol number 060/2011) and were in accordance with the National Institutes of Health guidelines.

## Additional files

### Major datasets

The following dataset was generated:

| Author(s) | Year | Dataset title | Dataset URL | Database, license, and accessibility information |
|---|---|---|---|---|
| Scheffer-Teixeira R, Tort A | 2016 | Multisite LFP recordings from the rat hippocampus during REM sleep | http://dx.doi.org/10.5061/dryad.12t21 | Available at Dryad Digital Repository under a CC0 Public Domain Dedication |

The following previously published dataset was used:

| Author(s) | Year | Dataset title | Dataset URL | Database, license, and accessibility information |
|---|---|---|---|---|
| Mizuseki K, Sirota A, Pastalkova E, Diba K, Buzsáki G | 2013 | Multiple single unit recordings from different rat hippocampal and entorhinal regions while the animals were performing multiple behavioral tasks | http://dx.doi.org/10.6080/K09G5JRZ | Publicly available at the Collaborative Research in Computational Neuroscience (http://crcns.org/) |

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
