## [Decision Letter]

[Editors’ note: a previous version of this study was rejected after peer review, but the authors submitted for reconsideration. The first decision letter after peer review is shown below.]

Thank you for submitting your work entitled "Lack of evidence for cross-frequency phase-phase coupling between theta and gamma oscillations in the hippocampus" for consideration by *eLife*. Your article has been favorably evaluated by Timothy Behrens (Senior Editor) and four reviewers, one of whom is a member of our Board of Reviewing Editors.

Our decision has been reached after consultation between the reviewers. Based on these discussions and the individual reviews, we regret to inform you that your work will not be considered further for publication in *eLife* at this time. Below we provide detailed feedback for your consideration if you choose to re-submit, and we encourage this possibility.

While all of the reviewers felt that this was an important and interesting paper, they also thought that it was presented in a confusing manner, was limited, and could lead to confusion rather than clarification for the field. Given the contrarian nature of the manuscript as presented, this would not be helpful. The reviewers' overall comments are provided here:

1) The paper presentation should be changed to be less confrontational and completely clear on what is being said and done. A well-crafted manuscript with clarity and sufficient explanations is lacking in the present submission.

A) Specifically, a more detailed review of the literature, a more careful presentation and discussion of the simulation results, and a more careful comparison with the procedures utilized in [Belluscio et al. 2012] is needed.

B) We note that phase-coupling is well defined theoretically from the tools of dynamical systems, but what is not clear is a selective measure of cross-frequency phase coupling. Defining an improved phase-phase coupling measure that detects true coupling and ignores artifactual coupling might be possible. Such a measure – even if it's not the optimal measure – might serve as a "patch" to existing approaches.

C) Finding the optimal phase-phase coupling measure (and assessing its statistical properties) would be a challenge. That is, a consistent and operational way to define how cross-frequency phase coupling can be measured (such that white noise or asymmetrical oscillation would not qualify). Clearly stating the lack of a clear measure as an open problem formally would be of value. In essence, the manuscript would need to make these aspects clear (i.e., present limitations of their work and others, possible fixes/challenges associated with having phase-phase coupling measures), and in this way can identify a methodological issue that's driving incorrect conclusions in the literature, and so be of service to the field.

2) Given that a clear measure of cross-frequency phase-coupling does not exist, a change of title should be considered.

3) A repository (of data and algorithms) should complement the work, and would be beneficial for the community moving forth as analyses, clear understanding and interpretation of artificial and biological data would be available.

[Editors’ note: what now follows is the decision letter after the authors submitted for further consideration.]

Thank you for submitting your article "On cross-frequency phase-phase coupling between theta and gamma oscillations in the hippocampus" for consideration by *eLife*. Your article has been favorably evaluated by Timothy Behrens (Senior Editor) and four reviewers, one of whom is a member of our Board of Reviewing Editors.

The reviewers have discussed the reviews with one another and the Reviewing Editor has drafted this decision to help you prepare a revised submission.

In this resubmitted paper on phase-phase coupling, all of the reviewers thought that several concerns had been addressed and suggestions implemented (such as the title change). The reviewers agreed that this is an important article that highlights a methodological issue impacting our understanding of the brain's activity and function. The authors are commended for sharing their analysis methods (on GitHub) and data.

The authors address three points in their paper: a statistical issue (using appropriates surrogate to detect n:m coupling), a conceptual issue (whether n:m coupling can be dissociated from asymmetrical oscillations), and an experimental one (whether n:m coupling is present in hippocampus). The first and last points are well made. However, this is not the case for the second, and other aspects remain unclear as presented. The authors are encouraged to revise and edit their paper to clarify all of the following issues.

1) A clear distinction between true phase versus estimated phase needs to be made as it is confusing as presented. Specifically:

A) As the authors' state, the theoretical quantity of n:m phase locking is well-defined. However, the estimated quantity of n:m phase locking as deduced from noisy neural data is fraught with difficulty. The authors show here that simply applying the theoretical n:m phase locking equation to data produces estimates that – without appropriate statistical tests – lead to spurious results. This is an important conclusion. Future work might seek to improve or correct (somehow) the estimation of the theoretical quantity, or propose a new theoretical quantity that is more easily estimated.

B) The point in the Discussion (subsection “Statistical inference of phase-phase coupling”, last paragraph) that true n:m coupling can be disentangled from asymmetrical oscillation by visual inspection of the LFP or EEG trace, precisely because they will look very much alike does not seem appropriate. This distinction may be hard to make at the macroscopic level (LFP/EEG) and could rather be searched for in how this signal is generated from the activity of distinct neural subpopulations. If the same gamma oscillations that are shown to phase-lock to theta oscillations can be dissociated neurally from theta oscillations (and indeed the authors did a tremendous job in previous publications at showing that hippocampal theta and gamma rely on distinct interconnected networks), then the evidence is rather of favor of genuine n:m locking. In the converse case, if no population selectively engaged in the faster oscillation can be identified, I would say that the asymmetrical oscillation is the most plausible explanation.

2) More detail of why the stripes in phase-phase plot can be produced by asymmetry of the theta waves needs to be provided.

It is not clear how the authors can conclude for the dataset at hand that stripes in phase-phase plot are due to the asymmetry of the theta waves and not to genuine gamma activity (Results, eighth paragraph). The authors seem to somehow tie this possible caveat to the stripes in phase-phase plots. However, stripes are just a visual inspection for maintained relationship between the two phases, just as (properly controlled) Rnm should measure.

3) The discussion about the caveat of asymmetrical oscillations is unsatisfying for the following reasons:

A) This point is made in the middle of the analysis of experimental data. Since this has nothing to do with the data at hand, in which no evidence for phase coupling was found, this obscures the message. A separate section should be devoted to the caveat of the asymmetrical oscillations, possibly before the analysis of experimental data.

B) The point should be clarified, by not only focussing on the presence of stripes for asymmetrical oscillations, but by explaining conceptually why this dissociation emerges. Possibly it could be illustrated by showing how an asymmetrical wave resembles the sum of multiples sinusoids at the fundamental and harmonics frequencies n:m locked to each other (and possibly with also some phase-amplitude coupling, see Kramer et al.).

4) Any differences between the measure of the authors and that of Belluscio's needs to be made clear. As specified by one of the reviewers:

It is not quite clear whether the methods used are exactly the same. In Belluscio et al., I think the counts were performed first for each surrogate (corresponding to a specific time-shift), and then the distribution of counts over the distinct time-shift was computed for each bin (i.e. Single Run analysis, which should be the correct method). By contrast, it seems that here – not quite clear in the text – surrogates for distinct time shifts were merged before counts were done in Figure 5 and Figure 4—figure supplement 1 (i.e. Pooled analysis, which of course averages the count and creates many false positives). [Note that I use Single Run vs. Pooled terminology to refer to whether the metrics was computed for each surrogate independently (Single Run) or for merged data, the metrics applying yo either Rnm or counts.] I hope that I am being wrong here. Otherwise that would seriously limit the conclusions of the author about invalidating Belluscio's findings.

Note that there is still a statistical problem with the Single Run analysis of phase-phase counts, which relates to the absence of a correction for the number of tested bins (either for a given theta phase or across all theta phases). In any case, the p-value in Belluscio et al. reaches very low value (some p<10^-10), which would probably resist proper corrections. If this is the case, that would provide statistically sound evidence for n:m coupling (asymmetrical theta being still a possible confound). Do the authors observe such low values in their own dataset? Could they remove statistical significance in the stripes in white noise using an appropriate correction? Or does this deceiving statistical significance come from yet another explanation?

[Editors' note: further revisions were requested prior to acceptance, as described below.]

Thank you for resubmitting your work entitled "On cross-frequency phase-phase coupling between theta and gamma oscillations in the hippocampus" for further consideration at *eLife*. Your revised article has been favorably evaluated by Timothy Behrens (Senior editor), a Reviewing editor, and three reviewers.

The manuscript has been improved but there are remaining issues that need to be addressed. All of the reviewers appreciated the authors' revisions and thought that this work could serve as a useful cautionary tale for neuroscientists as well as a useful starting point for continuing work. Even though it was felt that the results were not completely satisfying, it was also felt that it would be of great benefit in preventing researchers from wandering down the wrong analysis path. It was further noted that if this paper were presented as a statistics paper it could be improved more but then it would probably be less read and absorbed by the neuroscience community for some time.

An overall suggestion is that the authors should be more cautious in their interpretation and clearer about there being room for further progress in editing their manuscript.

Specific issues to address are:

1) The Abstract would benefit from significant revisions considering the various changes already done, and ones to be done. Sentences that should be targeted are: "filtered white noise has similar n:m phase-locking levels as actual data", and "the diagonal stripes in theta-gamma phase-phase histograms of actual data can be explained by theta harmonics".

2) Validity of Belluscio's analysis – overall conclusion:

A) In the rebuttal, the authors consider that the analysis by Belluscio is intrinsically flawed rather than being simply not well controlled statistically. I tend to defend the alternative option (clearly here I do not refer to the analysis of Mean Phase-Phase Plot, which the authors have convincingly showed that is flawed, but to the tests on phase counts; I also want to reassure the authors that I really do not have any personal motivation to go one way or the other). My intuition is that the presence of one single significant count in the phase-phase plot, if appropriately controlled for multiple comparisons, would provide a valid statistical measure. In essence it would not simply detect the presence of stripes (which surrogate run also feature) but measure whether their amplitude is larger than those of surrogates. This could be shown in a quite straightforward way by looking whether significant points in phase-phase plots obtained from white noise persist when a correction for multiple comparisons is applied. The authors seem to agree that they would not. (As for the type of correction, the Bonferroni method that the authors refer to looks too conservative as the bin counts clearly are non-independent; less stringent correction such as FDR or Holm-Bonferroni may be preferred). A count is not a metric for n:m coupling for sure, but it can inform of the particular concentration of the high frequency phase at one specific phase of the high frequency phase. More importantly, what matters here is not a metric of the coupling strength, but a reliable statistical test that selectively detects coupling, and my intuition is that counts in the phase-phase plot may provide one. If a simple correction can be applied to give a sound statistical test, this could allow Belluscio and colleagues to look back at their data and see whether the significant counts indeed resist correction for multiple comparisons.

B) About the very low p-values in Belluscio's data, the authors suggest that "it is possible that, by analyzing a longer epoch length, the influence of theta harmonics becomes more apparent and would lead to lower p-values, while the effect of the filtering-induced sinusoidality is washed out for both actual and surrogate epochs (of the same length)."

Well the manuscript previously demonstrates that n:m coupling measures as well as phase-phase plots cannot tear apart asymmetrical waves from true cross-frequency coupling. Thus, if indeed p-values remain lower than threshold when controlled appropriately, it could equally be due to asymmetrical theta or true theta-gamma phase coupling, but the latter could not be dismissed.

3) The reason for stripes in the phase-phase plot of hippocampal data:

A) In the manuscript authors state that "the diagonal stripes in phase-phase plots are due to theta harmonics and not to genuine gamma activity." If it were true harmonics, that would imply a consistent phase relationship between theta and gamma over sustained periods (not just over small periods as for white noise), i.e. would test positive for n:m coupling. So in my opinion here the effect is rather due to the temporary n:m alignment of phases, just as for white noise. In other words, stripes emerge also when analyzing white noise despite there being no harmonics in the signal.

B) In the rebuttal, the authors defend that "Believing in two different and genuine gamma activities – one coupled at 5 cycles per theta, the other at 4 cycles per theta – would go against the parsimonious principle." (this is when theta frequency evolves). However, as established by theoretical studies of coupled oscillators, n:m coupling is not a fixed property of the network but emerges as a combination of oscillator intrinsic dynamics and the characteristics of the connectivity pattern. See for example Arnold tongue: if we assume there is just one fixed gamma, a fluctuating theta and fixed connectivity, it is perfectly normal that n:m coupling will shift from 1:5 to 1:4 if the lower oscillator accelerates and the ratio of frequency goes from around 5 to around 4.

4) Asymmetrical theta vs. phase-coupled theta-gamma in general

I am coming back to the authors' comment on this in the rebuttal, although this is no longer present in the manuscript. This is a comment for the authors benefit and speaks to the overall comment mentioned above. Sharp edges in LFP/EEG are not by itself an indication that we are measuring a single asymmetrical oscillation as the superposition of a slow and a fast n:m coupled oscillation can also give rise to sharp edges (Figure 4; note that this can even be obtained with just two oscillations). Thus, sharp edges is no more a selective feature of asymmetrical oscillations than n:m coherence is selective of phase-coupled oscillators. In other words, visual inspection cannot tell us more than statistics.

5) It looks like "Random Perm" is more powerful test than "Time Shift", as mentioned in the first paragraph of the subsection “Lack of evidence for n:m phase-locking in actual LFPs”. If this is the case then it could be stated as a conclusion of the work that "Random perm" should be preferred over "Time Shift" (and of course of the "Scrambling" procedure), as it is less likely to miss existing effects (lower False Rejections rate).

6) Conclusion: the very new paper by Lozano-Soldevilla and colleagues (Frontiers Comp Neuro) provides another example of spurious cross-frequency coupling measures due to asymmetrical oscillations, could be worth referencing.

---

## [Author Response]

[Editors’ note: the author responses to the first round of peer review follow.]

While all of the reviewers felt that this was an important and interesting paper, they also thought that it was presented in a confusing manner, was limited, and could lead to confusion rather than clarification for the field. Given the contrarian nature of the manuscript as presented, this would not be helpful. The reviewers' overall comments are provided here:

*1) The paper presentation should be changed to be less confrontational and completely clear on what is being said and done. A well-crafted manuscript with clarity and sufficient explanations is lacking in the present submission.*

We have made several modifications in our manuscript to let it less confrontational, including a change in the title. We have also edited our Results section to clarify points raised by reviewer 3.

*A) Specifically, a more detailed review of the literature, a more careful presentation and discussion of the simulation results, and a more careful comparison with the procedures utilized in [Belluscio et al. 2012] is needed.*

We have included new references mentioned by the reviewers, and performed new analyses and simulations to address important comments by reviewer 3; this led to the elaboration of new figures and text. To further conform to the reviewer’s comments, we have deleted former Figure 1 and related text, and restructured the way we present and discuss our results. Finally, we now provide p- value maps to allow straight comparison with the procedure performed in Belluscio et al. (2012).

*B) We note that phase-coupling is well defined theoretically from the tools of dynamical systems, but what is not clear is a selective measure of cross-frequency phase coupling. Defining an improved phase-phase coupling measure that detects true coupling and ignores artifactual coupling might be possible. Such a measure – even if it's not the optimal measure – might serve as a "patch" to existing approaches.*

*C) Finding the optimal phase-phase coupling measure (and assessing its statistical properties) would be a challenge. That is, a consistent and operational way to define how cross-frequency phase coupling can be measured (such that white noise or asymmetrical oscillation would not qualify). Clearly stating the lack of a clear measure as an open problem formally would be of value. In essence, the manuscript would need to make these aspects clear (i.e., present limitations of their work and others, possible fixes/challenges associated with having phase-phase coupling measures), and in this way can identify a methodological issue that's driving incorrect conclusions in the literature, and so be of service to the field.*

We will answer to points B and C together:

The definition of cross-frequency phase coupling follows the exact same rationale as the definition of phase-coupling, and the n:m phase-coupling metric is the natural expansion of the 1:1 phase- coupling metric. We believe both metrics are well defined. The current concept of n:m phase-locking reflected in the metric’s definition (i.e., the concentration of the phase difference distribution) is used not only by experimentalists but also by theoretical groups. Please note that we did not create the n:m phase-locking metric in this work but studied the exact same metric in current widespread use in the literature, which led influential conclusions about how the brain might work. In our paper, we identify the lack of proper control/surrogate analyses as the methodological issue driving incorrect conclusions, which should prompt other groups to revisittheir findings. We believe these results are of enough importance to many neuroscientists working with brain rhythms and their interactions (e.g., Schack et al., 2002; Schack et al., 2005; Schack and Weiss, 2005; Palva et al., 2005; Holz et al., 2010; Sauseng et al., 2008; Sauseng et al., 2009; Zheng et al., 2013; Xu et al., 2013; Stujenske et al., 2014; Xu et al., 2015; Aru et al., 2015; Hyafill et al., 2015; Chaieb et al., 2015; Zheng et al., 2016, among many others).

The points above seem to implicitly assume that n:m phase-phase coupling would exist in the brain but the current metric is not able to detect it. While we have difficulties in imagining how this would exactly happen (since the metric and the definition of n:m phase coupling are intrinsically related), we do not disagree with such possibility and have edited our text to discuss it. In addition, we performed new simulations to show that cases of significant coupling may reflect not only true coupling but also artifactual coupling. We tried but were unable to devise a way to separate these cases; in the revised version, we have expanded our Results and Discussion sections to comment on these limitations.

But please note that our results point to lack of statistically significant coupling in actual LFPs upon proper surrogate procedures, and therefore such issues do not interfere with the main conclusion of our paper.

*2) Given that a clear measure of cross-frequency phase-coupling does not exist, a change of title should be considered.*

We have changed our title to “On cross-frequency phase-phase coupling between theta and gamma oscillations in the hippocampus”. But please note that above and in our answers below we defend that there exists a current, well-defined measure of n:m phase-locking, and that the lack of evidence for n:m phase-coupling reported in our work may not be due to a “fault” of this metric but simply to lack of existence of such effect.

*3) A repository (of data and algorithms) should complement the work, and would be beneficial for the community moving forth as analyses, clear understanding and interpretation of artificial and biological data would be available.*

We have made our codes available at https://github.com/tortlab/phase_phase. In addition, we also made the analyzed data available at Dryad (http://datadryad.org/). If the manuscript is accepted for publication, the dataset will have the doi: 10.5061/dryad.12t21.

[Editors' note: the author responses to the re-review follow.]

*In this resubmitted paper on phase-phase coupling, all of the reviewers thought that several concerns had been addressed and suggestions implemented (such as the title change). The reviewers agreed that this is an important article that highlights a methodological issue impacting our understanding of the brain's activity and function. The authors are commended for sharing their analysis methods (on GitHub) and data.*

*The authors address three points in their paper: a statistical issue (using appropriates surrogate to detect n:m coupling), a conceptual issue (whether n:m coupling can be dissociated from asymmetrical oscillations), and an experimental one (whether n:m coupling is present in hippocampus). The first and last points are well made. However, this is not the case for the second, and other aspects remain unclear as presented. The authors are encouraged to revise and edit their paper to clarify all of the following issues.*

*1) A clear distinction between true phase versus estimated phase needs to be made as it is confusing as presented. Specifically:*

*A) As the authors' state, the theoretical quantity of n:m phase locking is well-defined. However, the estimated quantity of n:m phase locking as deduced from noisy neural data is fraught with difficulty. The authors show here that simply applying the theoretical n:m phase locking equation to data produces estimates that – without appropriate statistical tests – lead to spurious results. This is an important conclusion. Future work might seek to improve or correct (somehow) the estimation of the theoretical quantity, or propose a new theoretical quantity that is more easily estimated.*

We understand the reviewer’s point: for any given signal the n:m phase-locking metric will provide an estimated value irrespective of the existence of true phase-phase coupling or not; the former can only be inferred when comparing the estimated value with a chance/surrogate distribution. But we note that this issue is not specific for n:m phase-locking estimates, as it also occurs for other metrics commonly used in Neuroscience. For instance, estimates of mutual information, coherence, spike-field coupling, phase-amplitude coupling, directionally measures – to mention a few – all suffer from short sample bias. In the re-revised manuscript, we now devote a paragraph to discuss such issue and highlight the difference between estimated and true phase-phase coupling (please see subsection “Statistical inference of phase-phase coupling”, second paragraph).

*B) The point in the Discussion (subsection “Statistical inference of phase-phase coupling”, last paragraph) that true n:m coupling can be disentangled from asymmetrical oscillation by visual inspection of the LFP or EEG trace, precisely because they will look very much alike does not seem appropriate. This distinction may be hard to make at the macroscopic level (LFP/EEG) and could rather be searched for in how this signal is generated from the activity of distinct neural subpopulations. If the same gamma oscillations that are shown to phase-lock to theta oscillations can be dissociated neurally from theta oscillations (and indeed the authors did a tremendous job in previous publications at showing that hippocampal theta and gamma rely on distinct interconnected networks), then the evidence is rather of favor of genuine n:m locking. In the converse case, if no population selectively engaged in the faster oscillation can be identified, I would say that the asymmetrical oscillation is the most plausible explanation.*

The passage of our Discussion mentioned by the reviewer reads, former version):

“To the best of our knowledge, there is currently no metric capable of automatically distinguishing true cross-frequency coupling from waveform-induced artifacts in collective signals such as LFP, EEG and MEG signals. Until one is developed, we believe the best practice is to judge statistically significant coupling levels on a case-by-case basis by visual inspection of raw signals.”

We meant that if theta and gamma oscillations genuinely n:m phase-couple, one should be able to observe this effect in the unfiltered LFP (i.e., one should see the actual gamma cycles coupled to theta upon visual inspection); on the other hand, if n:m coupling is spurious and due to an asymmetric theta wave, one would observe sharp deflections in the unfiltered LFP instead of genuine gamma cycles. We have previously discussed such “manual” solution in Kramer et al. (2008) for the cases of spurious phase-amplitude coupling. On page 356 of that paper, we wrote:

“To ameliorate the potential problem of spurious frequency comodulation, we recommend careful inspection of the unfiltered data to distinguish between sharp edges and true high frequency oscillations.” And later: “First, we recommend visual inspection of the unfiltered data at times corresponding to increases in the amplitude envelope of the high frequency activity. At these times, do the unfiltered data exhibit high frequency oscillations or sharp edges?”

But we acknowledge that a problem arises when the signal has both gamma oscillations and sharp deflections, in which case it is very difficult – if not impossible – to realize by visual inspection which effect is at play, and therefore we agree with the reviewer that the sentence may not be appropriate.

We further agree with the reviewer that understanding how the signal is generated by the activity of distinct neural subpopulations would be ideal to sort apart true from artifactual coupling. But this constitutes a huge problem in Neuroscience by itself: after all, what gives rise to LFP oscillations? After decades of research, we still debate about what generates theta and gamma at the circuitry and neuronal levels; we currently have models but not a complete understanding of oscillatory genesis; the available experimental data are not yet fully conclusive. Computer simulations – as the ones mentioned by the reviewer – are useful tools to generate insights into potential mechanisms, but they cannot be considered ultimate proofs. Thus, the gold standard solution referred to by the reviewer (of searching how the signal is generated by the activity of neural subpopulations) is unfortunately not a practical one but methodologically very challenging.

To address this issue, in the re-revised version we removed the sentence the reviewer did not find appropriate, and instead we now spell out the ideal suggestion put forward by the reviewer (subsection “Statistical inference of phase-phase coupling”, last paragraph).

*2) More detail of why the stripes in phase-phase plot can be produced by asymmetry of the theta waves needs to be provided.*

*It is not clear how the authors can conclude for the dataset at hand that stripes in phase-phase plot are due to the asymmetry of the theta waves and not to genuine gamma activity (Results, eighth paragraph). The authors seem to somehow tie this possible caveat to the stripes in phase-phase plots. However, stripes are just a visual inspection for maintained relationship between the two phases, just as (properly controlled) Rnm should measure.*

We agree with the reviewer that stripes in phase-phase plots are only a visual manifestation of a maintained phase relationship, and in essence the stripes reflect what R_n:m_ measures. The stripes may be due to (1) theta harmonics (i.e., theta asymmetry), due to (2) genuinely coupled gamma activity, or even due to (3) the sinusoidality imposed by the filter. Similarly, all these 3 effects can lead to a bump in the R_n:m_ curve (see Figure 3 for genuine coupling, Figure 4 for theta asymmetry, and Figure 2 and Figure 2—figure supplement 1 for the filtering-imposed sinusoidality in white noise).

Therefore, the reviewer is right in that it is not possible to differentiate genuinely coupled gamma from theta harmonics (or filtering effects) solely based on visual inspection of phase-phase plots, and the same is also true for the visual inspection of a bump in the R_n:m_ curve. But please note that we did not mean to imply that the presence of stripes per se was a proof of theta harmonics and absence of genuine coupling. Rather, our conclusion of absence of genuine coupling stemmed from a conjunction of observations, along with the Occam's principle of parsimony:

1) As correctly pointed out by the reviewer, the R_n:m_ metric and the stripes are essentially related: the more prominent the stripes are in phase-phase plots, the higher the R_n:m_. Therefore, were the stripes due to genuinely coupled gamma activity, they should be associated with R_n:m_ values different from chance, which was not the case (Figure 5 and its figure supplements).

2) As shown in Figure 7, the number of stripes perfectly relates to the order of the first theta harmonics falling within the filtered bandwidth. That is, instead of assuming the existence of multiple types of gamma activities coupled to theta, we deemed more parsimonious to attribute the different number of stripes shown in Figure 7 to different orders of the theta harmonics.

3) Please notice in Figure 8 that the filter bandwidth for gamma is the same (30-50 Hz). Yet, the phase-phase plot of the LFP in 8A has 5 stripes while the phase-phase plot of the LFP in 8B has 4 stripes. Instead of assuming that theta would couple to a genuine 1:5 gamma (i.e., 5 gamma cycles per theta cycle) and to a genuine 1:4 gamma (4 gamma cycles per theta cycle), we believe the stripes are straightforwardly explained by the fact that theta frequency in 8A example is 7.1 Hz while in 8B theta frequency is 8.4 Hz; therefore, it takes a factor of 5 and 4, respectively, to make these theta frequencies larger than the low cutoff frequency of the gamma filter (i.e., 7.1 Hz x 5 = 35.5 Hz > 30 Hz and 8.4 x 4 = 33.6 Hz > 30 Hz), which would explain why we see 5 and 4 stripes, respectively. Believing in two different and genuine gamma activities – one coupled at 5 cycles per theta, the other at 4 cycles per theta – would go against the parsimonious principle.

But while our observations argue against genuine gamma coupling, the distinction between the contribution of theta harmonics and of filtering-induced sinusoidality is trickier, and actually both effects may be at play. In brief, though, the effect of filtering-induced sinusoidality is short lived and would not account for the stripes seen in very long epochs such as the 20-min long signal analyzed in Figure 7.

In the re-revised manuscript, we elaborated new text to clarify these points (subsection “Statistical inference of phase-phase coupling”, fourth paragraph).

*3) The discussion about the caveat of asymmetrical oscillations is unsatisfying for the following reasons:*

*A) This point is made in the middle of the analysis of experimental data. Since this has nothing to do with the data at hand, in which no evidence for phase coupling was found, this obscures the message. A separate section should be devoted to the caveat of the asymmetrical oscillations, possibly before the analysis of experimental data.*

We thank the reviewer for the nice suggestion, which we have followed in the re-revised version. Namely, we now divide our Results section into subsections (in the former version we had a single text), and devote one subsection – before we present the experimental results – to show the caveat of asymmetrical/non-sinusoidal oscillations. This has led to major changes throughout the manuscript, such as the elaboration of a new figure and associated text (see below), and re-ordering of previous figures and text. For convenience, we submitted as “related manuscript file” a PDF version of our manuscript with all changes highlighted.

*B) The point should be clarified, by not only focussing on the presence of stripes for asymmetrical oscillations, but by explaining conceptually why this dissociation emerges. Possibly it could be illustrated by showing how an asymmetrical wave resembles the sum of multiples sinusoids at the fundamental and harmonics frequencies n:m locked to each other (and possibly with also some phase-amplitude coupling, see Kramer et al.).*

To address this point, we have elaborated a new figure in which we illustrate that the Fourier decomposition of a non-sinusoidal wave is characterized by the sum of multiple sinusoids at harmonic frequencies that are n:m phase-locked to each other; please see new Figure 4 and new subsection of the Results section (c.f. our answer above). We mention that an asymmetric wave shape may also lead to spurious phase-amplitude coupling in the first paragraph of the subsection “Spurious n:m phase-locking due to non-sinusoidal waveforms” and in the last paragraph of the subsection “Statistical inference of phase-phase coupling”, and explicitly show this effect in new Figure 4—figure supplement 1.

*4) Any differences between the measure of the authors and that of Belluscio's needs to be made clear. As specified by one of the reviewers:*

*It is not quite clear whether the methods used are exactly the same. In Belluscio et al., I think the counts were performed first for each surrogate (corresponding to a specific time-shift), and then the distribution of counts over the distinct time-shift was computed for each bin (i.e. Single Run analysis, which should be the correct method). By contrast, it seems that here – not quite clear in the text – surrogates for distinct time shifts were merged before counts were done in Figure 5 and Figure 4—figure supplement 1 (i.e. Pooled analysis, which of course averages the count and creates many false positives). [Note that I use Single Run vs. Pooled terminology to refer to whether the metrics was computed for each surrogate independently (Single Run) or for merged data, the metrics applying yo either Rnm or counts.] I hope that I am being wrong here. Otherwise that would seriously limit the conclusions of the author about invalidating Belluscio's findings.*

The statistical analysis of the phase-phase plots was done exactly as described in Belluscio et al. and correctly summarized by the reviewer. Namely, for each phase-phase bin, each single surrogate run contributed with one count (please note that we did not merge single surrogates before counting). Then, for each phase-phase bin we constructed a distribution of counts, and, based on the mean and standard deviation of this distribution, we inferred the statistical significance of the actual count for that bin in the phase-phase plot.

In the re-revised version, we have edited the legend and panel labels of Figure 7, Figure 7—figure supplement 1, Figure 8, Figure 8—figure supplement 1 (current numeration) and subsection “Lack of evidence for n:m phase-locking in actual LFPs”, second paragraph and subsection “Statistical inference of phase-phase coupling”, third paragraph to let it clearer that the mean and SD are computed over individual bin counts, and not over pooled counts.

Importantly, please note that we do not consider the framework employed in Belluscio et al. as a Single Run analysis. We reserve the term “Single Run” to denote an analysis in which each single surrogate run generates a single measure of n:m phase-locking. In this sense, we don’t see a single histogram count as a metric of n:m phase-locking, in the same way that we don’t see a single vector of phase difference (ei�φnm(t)) as a metric of n:m phase-locking, simply because neither a single histogram count nor a single phase difference vector measures the level of n:m phase-locking between the time series. Rather, for each single surrogate run, a metric of n:m phase-locking is obtained by computing the length of the mean vector over all phase difference vectors (R_n:m_). Therefore, sticking to this analogy, the counts in phase-phase plots would correspond to the phase difference vectors used to compute the R_n:m_, but not to the R_n:m_ metric itself. Again, a count is not an n:m phase-locking metric. In the case of the phase-phase plot, a Single Run surrogate would be an entire phase-phase plot constructed from a single surrogate (i.e., time-shifted) time series, because it is at this level that we infer the existence of n:m phase-locking – and not at the level of a single bin count. In other words, Belluscio et al. inferred the existence of n:m phase-locking by the visual inspection of stripes. And, as shown in Figure 8—figure supplement 1, phase-phase plots from single surrogate runs do exhibit stripes similar to the original phase-phase plot. This relates to the fact that single surrogate R_n:m_ values are similar to original R_n:m_ values, that is, there is no difference between actual and surrogate data at the level of an n:m phase-locking metric, which justifies our conclusions.

In the re-revised version, we substantially edited a previous Discussion paragraph to clarify these points (please see subsection “Statistical inference of phase-phase coupling”, third paragraph).

Supplementary remarks: In the right panel of Figure 6, Belluscio et al. show the mean phase-phase plot over all surrogate runs. In our paper we note that plotting the mean number of counts across all surrogates is qualitatively identical to plotting the pool of counts over all surrogate runs, up to a scaling factor, as shown in Figure 11.

Author response image 1.**DOI:**
http://dx.doi.org/10.7554/eLife.20515.031

Clearly, the scaling factor is the number of surrogates (n=1000). The equivalence above motivated us to state that the procedure in Belluscio et al. is akin to what we call as a pooled surrogate analysis: there is no visual inspection of phase-phase plots for single surrogate runs but only the inspection of their average, which is equivalent to the visual inspection of their pool (of note, this improper visual analysis was also performed in Zheng et al. 2016). At any event, we stress that there is no proper Single Run surrogate analysis performed at the level of an n:m phase-locking metric such as the R_n:m_ in Belluscio et al.

*Note that there is still a statistical problem with the Single Run analysis of phase-phase counts, which relates to the absence of a correction for the number of tested bins (either for a given theta phase or across all theta phases). In any case, the p-value in Belluscio et al. reaches very low value (some p<10^-10), which would probably resist proper corrections. If this is the case, that would provide statistically sound evidence for n:m coupling (asymmetrical theta being still a possible confound). Do the authors observe such low values in their own dataset? Could they remove statistical significance in the stripes in white noise using an appropriate correction? Or does this deceiving statistical significance come from yet another explanation?*

Indeed, in most phase-phase bins the counts would not survive a Bonferroni correction. (Since we used 120x120 phase-phase bins, a correction would bring the significance threshold down to 10^-6^; we don’t know how many phase-phase bins Belluscio et al. employed and whether their counts would survive a Bonferroni correction). But please notice that this is actually one more argument against the existence of phase-phase coupling. Forgetting for the moment the issues with this framework mentioned above, were the described effect robust, we should have been able to replicate it.

We suspect the p-values in Belluscio et al. are lower than ours due to differences in epoch length; they stated they analyzed 11 sessions in three rats, but the total analyzed length is not informed. Intuitively, we believe it is possible that, by analyzing a longer epoch length, the influence of theta harmonics becomes more apparent and would lead to lower p-values, while the effect of the filtering-induced sinusoidality is washed out for both actual and surrogate epochs (of the same length).

As for this statement: “If this is the case, that would provide statistically sound evidence for n:m coupling (asymmetrical theta being still a possible confound).”

As reviewers ourselves, we appreciate the voluntary and time-consuming work of the reviewer, and thank him/her for the relevant suggestions that helped us improving our paper. But while we respect the reviewer’s opinion, we feel compelled to disagree with this particular statement. Based on arguments provided above, we do not think that a proper statistical procedure was performed in Belluscio et al. After all, what is the n:m phase-locking metric being measured in Belluscio et al. framework? A single phase-phase bin count is not a metric, it does not inform how coupled the time series are, and even coupled oscillators have non-significant bin counts. Please note that Belluscio et al. state “The diagonal stripes in the phase–phase plots indicate a statistically reliable relationship between the respective phases of theta and gamma oscillations during both RUN and REM”.Thus, the “metric” in Belluscio et al. is in essence the visual inspection of stripes. But single run surrogates also exhibit stripes (Figure 8—figure supplement 1), in the same way that single run surrogates also exhibit a bump in the R_n:m_ curve. The stripes are trivially averaged out when one computes the mean over time-shifted phase-phase plots, which relates to a pooled analysis (c.f. Figure 11); it is not too surprising that actual counts of some phase-phase bins are deemed statistically significant under this framework. Please note that a single surrogate phase-phase plot – which has stripes – would also be deemed statistically significant if one were to apply this very same framework. Therefore, we do not consider this framework a proper one. Instead, Belluscio et al. should have inferred the statistical significance of coupling at the R_n:m_ level, which is an actual metric of n:m phase-locking.

In the re-revised version, we have substantially edited a previous Discussion paragraph to explain why an histogram count would not constitute an n:m phase-locking metric, and why we do not consider the analysis framework in Belluscio et al. a proper one. Please see subsection “Statistical inference of phase-phase coupling”, third paragraph.

[Editors' note: further revisions were requested prior to acceptance, as described below.]

*[…] An overall suggestion is that the authors should be more cautious in their interpretation and clearer about there being room for further progress in editing their manuscript.*

*Specific issues to address are:*

*1) The Abstract would benefit from significant revisions considering the various changes already done, and ones to be done. Sentences that should be targeted are: "filtered white noise has similar n:m phase-locking levels as actual data", and "the diagonal stripes in theta-gamma phase-phase histograms of actual datacan be explained by theta harmonics".*

We thank the reviewer for calling attention to this issue. In the revised version we have edited the Abstract to reflect changes done in the previous and current version. The two mentioned sentences no longer appear in the revised version.

*2) Validity of Belluscio's analysis – overall conclusion:*

*A) In the rebuttal, the authors consider that the analysis by Belluscio is intrinsically flawed rather than being simply not well controlled statistically. I tend to defend the alternative option (clearly here I do not refer to the analysis of Mean Phase-Phase Plot, which the authors have convincingly showed that is flawed, but to the tests on phase counts; I also want to reassure the authors that I really do not have any personal motivation to go one way or the other). My intuition is that the presence of one single significant count in the phase-phase plot, if appropriately controlled for multiple comparisons, would provide a valid statistical measure. In essence it would not simply detect the presence of stripes (which surrogate run also feature) but measure whether their amplitude is larger than those of surrogates. This could be shown in a quite straightforward way by looking whether significant points in phase-phase plots obtained from white noise persist when a correction for multiple comparisons is applied. The authors seem to agree that they would not. (As for the type of correction, the Bonferroni method that the authors refer to looks too conservative as the bin counts clearly are non-independent; less stringent correction such as FDR or Holm-Bonferroni may be preferred). A count is not a metric for n:m coupling for sure, but it can inform of the particular concentration of the high frequency phase at one specific phase of the high frequency phase. More importantly, what matters here is not a metric of the coupling strength, but a reliable statistical test that selectively detects coupling, and my intuition is that counts in the phase-phase plot may provide one. If a simple correction can be applied to give a sound statistical test, this could allow Belluscio and colleagues to look back at their data and see whether the significant counts indeed resist correction for multiple comparisons.*

Indeed, we tended not to consider Belluscio et al. analysis a proper one; despite a bin count not being a metric of n:m coupling strength (c.f. our previous answers and acknowledged by the reviewer), we predicted difficulties in interpreting significant bin counts. For instance, a phase-phase plot of 120 x 120 bins (i.e., 3^o^ bins) has 14400 bin counts. If a single bin count out of these 14400 bins is deemed statistically significant (after an appropriate correction), should we consider the time series n:m phase-coupled? We particularly did not consider that having only one bin deemed statistically significant would suffice. But then, what would be the minimal number of significant bins for assuming n:m phase-coupling? And would the appearance of striped patterns be necessary? Or would a few significant bins scattered at random locations suffice? Again, especially considering that there is a well-defined metric of n:m phase-coupling (R_n:m_), we had particular difficulty in considering the bin count analysis a proper one. That said, though, we do respect the opposite opinion manifested by the reviewer, and have followed the suggestion of subjecting the p-values of phase-phase bin counts to a correction for multiple comparisons. These analyses show that the statistically significant bins in phase-phase plots of white noise under the original framework of Belluscio et al. do not persist after correcting for multiple comparisons using Holm-Bonferroni (we considered the FDR correction too liberal since it led to significant bins in white noise). The results were true when building surrogate distribution using both Time Shift and Random Permutation procedures. This is now shown in new Figure 9 and stated/discussed in the fourth paragraph of the subsection “On diagonal stripes in phase-phase plots” and in the third paragraph of the subsection “Statistical inference of phase-phase coupling”. Please note that we created a new Results subsection to discuss the origin of stripes in phase-phase plots (“On diagonal stripes in phase-phase plots”), which led to editions and rearrangement of previous paragraphs and figures (in specific, former Figure 5, Figure 6 and Figure 7 are now new Figure 6, Figure 7 and Figure 8, respectively).

In actual LFP data, we also found no significant bins after correcting for multiple comparisons when employing the same ± 200 ms time-shift surrogate procedure as in Belluscio et al. Interestingly, however, for very long epochs (5-10 minutes of concatenated data) we did find significant bins even after Holm-Bonferroni correction when employing randomly permutated – but not time-shifted – surrogates. We believe this is because small time-shifts as performed in Belluscio et al. ( ± 200 ms) would preserve phase-locking between the fundamental and harmonic frequencies (albeit in different phase relations) while the larger time shifts obtained through random permutations would not. These results are now shown in new Figure 10 stated/discussed in the last two paragraphs of the subsection “On diagonal stripes in phase-phase plots” and in the last paragraph of the subsection “Statistical inference of phase-phase coupling”. Please note that in the revised manuscript we leave room for this result being due to a weak but true coupling effect (subsection “On diagonal stripes in phase-phase plots”, fifth paragraph, subsection “Statistical inference of phase-phase coupling”, last paragraph, and subsection “Lack of evidence vs evidence of non-existence”, last paragraph). Please see also our answers below.

*B) About the very low p-values in Belluscio's data, the authors suggest that "it is possible that, by analyzing a longer epoch length, the influence of theta harmonics becomes more apparent and would lead to lower p-values, while the effect of the filtering-induced sinusoidality is washed out for both actual and surrogate epochs (of the same length)."*

*Well the manuscript previously demonstrates that n:m coupling measures as well as phase-phase plots cannot tear apart asymmetrical waves from true cross-frequency coupling. Thus, if indeed p-values remain lower than threshold when controlled appropriately, it could equally be due to asymmetrical theta or true theta-gamma phase coupling, but the latter could not be dismissed.*

As stated above and shown in new Figure 10, p-values in phase-phase plots of actual data are no longer considered statistically significant after appropriate correction for multiple comparisons when employing the surrogate procedure introduced in Belluscio et al. ( ± 200 ms time shifts), which in principle would settle this issue. However, we note that bin counts are deemed statistically significant even after correction when employing randomly permutated surrogates, especially for very long epochs (Figure 10). We agree with the reviewer that n:m coupling measures cannot separate true coupling from asymmetrical waves; while we indeed tend to ascribe these results to harmonic influences, in the revised manuscript we now state our interpretation more cautiously and leave open the possibility of true theta-gamma phase coupling (subsection “On diagonal stripes in phase-phase plots”, fifth paragraph, and the Conclusion).

*3) The reason for stripes in the phase-phase plot of hippocampal data:*

*A) In the manuscript authors state that "the diagonal stripes in phase-phase plots are due to theta harmonics and not to genuine gamma activity." If it were true harmonics, that would imply a consistent phase relationship between theta and gamma over sustained periods (not just over small periods as for white noise), i.e. would test positive for n:m coupling. So in my opinion here the effect is rather due to the temporary n:m alignment of phases, just as for white noise. In other words, stripes emerge also when analyzing white noise despite there being no harmonics in the signal.*

We thank the reviewer for the insightful comment. Indeed, after performing new analyses on white noise and actual LFPs (which led to new Figure 9, Figure 10, Figure 10—figure supplement 1 and Figure 10—figure supplement 2), we came to agree with the reviewer that the temporary n:m phase alignment due to the sinusoidality induced by filtering is a main factor in determining the diagonal stripes in phase-phase plots. These conclusions are backed by data shown in our new figures and explicitly stated in subsection “On diagonal stripes in phase-phase plots”, last three paragraphs and subsection “Statistical inference of phase-phase coupling”, last paragraph. However, our new analyses now reveal that in actual LFP signals there is a second influence, which is not accounted for solely by filtering. This effect is only statistically detected when analyzing very long epoch lengths (>100 s) and when contrasting original results (either phase-phase plots or Rnm curves) against randomly permutated data (please see new Figure 10). Based on our analysis of sawtooth wave (in which random permutation was more sensitive to detect coupling than time-shifted surrogates, Figure 4—figure supplement 2), and given that the effect of filtering-induced sinusoidality is controlled for in surrogate data, we particularly believe that the weak (Rnm=0.03) but significant coupling detected in very long time series is due to theta harmonics (discussed in subsection “Statistical inference of phase-phase coupling”, last paragraph). But in the revised manuscript we leave open the possibility of this low level of coupling as denoting true coupling (subsection “On diagonal stripes in phase-phase plots”, fifth paragraph, subsection “Statistical inference of phase-phase coupling”, last paragraph, subsection “Lack of evidence vs. evidence of non-existence”, last paragraph, and the Conclusion).

*B) In the rebuttal, the authors defend that "Believing in two different and genuine gamma activities – one coupled at 5 cycles per theta, the other at 4 cycles per theta – would go against the parsimonious principle." (this is when theta frequency evolves). However, as established by theoretical studies of coupled oscillators, n:m coupling is not a fixed property of the network but emerges as a combination of oscillator intrinsic dynamics and the characteristics of the connectivity pattern. See for example Arnold tongue: if we assume there is just one fixed gamma, a fluctuating theta and fixed connectivity, it is perfectly normal that n:m coupling will shift from 1:5 to 1:4 if the lower oscillator accelerates and the ratio of frequency goes from around 5 to around 4.*

We thank the reviewer for calling our attention to these facts, point taken. We just note that the “parsimony argument” properly criticized by the reviewer was not present in the manuscript text but only stated in the rebuttal letter.

*4) Asymmetrical theta vs. phase-coupled theta-gamma in general*

*I am coming back to the authors' comment on this in the rebuttal, although this is no longer present in the manuscript. This is a comment for the authors benefit and speaks to the overall comment mentioned above. Sharp edges in LFP/EEG are not by itself an indication that we are measuring a single asymmetrical oscillation as the superposition of a slow and a fast n:m coupled oscillation can also give rise to sharp edges (Figure 4; note that this can even be obtained with just two oscillations). Thus, sharp edges is no more a selective feature of asymmetrical oscillations than n:m coherence is selective of phase-coupled oscillators. In other words, visual inspection cannot tell us more than statistics.*

We appreciate the thoughtful comments by the reviewer. (As stated by the reviewer, this issue is no longer present in our manuscript.)

*5) It looks like "Random Perm" is more powerful test than "Time Shift", as mentioned in the first paragraph of the subsection “Lack of evidence for n:m phase-locking in actual LFPs”. If this is the case then it could be stated as a conclusion of the work that "Random perm" should be preferred over "Time Shift" (and of course of the "Scrambling" procedure), as it is less likely to miss existing effects (lower False Rejections rate).*

We agree (and actually our new analyses of very long LFP epochs support such conclusion). This conclusion is now explicitly stated in the last paragraph of the subsection “Statistical inference of phase-phase coupling”.

*6) Conclusion: the very new paper by Lozano-Soldevilla and colleagues (Frontiers Comp Neuro) provides another example of spurious cross-frequency coupling measures due to asymmetrical oscillations, could be worth referencing.*

We now cite Lozano-Soldevilla et al. et al. at the passage mentioned by the reviewer (currently in the last paragraph of the subsection “Statistical inference of phase-phase coupling”).